# Beyond Accuracy: Latent Perturbations for Cognitive-Aware Diagnosis

Yuting Yan [1]   Yinghao Fu [2 1]   Wendi Ren [1]   Haozhou Gao [1]   Shuang Li [1 *]

## Abstract

Diagnosing rare diseases remains a persistent challenge, often hindered by *cognitive anchoring*: once clinicians settle on a common diagnosis, they often discount alternative explanations, including rare conditions. To address this, we propose a cognitive-aware counterfactual reasoning framework using a Denoising Masked AutoEncoder (DMAE) to simulate *what-if* diagnostic scenarios that probe clinicians' initial assumptions. Our model jointly learns (1) the true distribution of diseases and symptoms, and (2) human diagnostic behavior, revealing critical gaps between *medically possible* and *clinically considered* diagnoses. By strategically perturbing latent patient representations, it generates *contrastive counterfactuals* that highlight rare yet plausible diseases that cognitive bias often obscures. Unlike traditional decision-support tools, our system suggests rare diseases not because they are statistically dominant, but because they are *systematically underconsidered* relative to the observed evidence and learned diagnostic behavior. Across four public and three private rare-disease datasets, our approach outperforms standard machine learning classifiers in detecting rare conditions while maintaining strong performance on common diagnoses. Beyond boosting accuracy, the counterfactual evidence encourages *hypothesis-driven reasoning* and supports clinical learning.

## 1. Introduction

Despite advances in machine learning for clinical diagnosis, *rare diseases remain notoriously difficult to identify* due to their low prevalence, heterogeneous manifestations, and frequent overlap with more common conditions (Schieppati et al., 2008; Griggs et al., 2009). Consider a patient presenting with persistent fatigue, joint pain, and skin rashes, clinicians often anchor on familiar diagnoses like lupus rather than considering rare alternatives such as Ehlers-Danlos syndrome. This diagnostic misdirection is not merely a result of statistical rarity or symptom ambiguity, but also due to a well-documented *cognitive bias* known as *anchoring*, in which clinicians settle prematurely on an initial diagnosis and revise it insufficiently when new or contradictory evidence emerges (Tversky & Kahneman, 1974; Saposnik et al., 2016; Croskerry, 2002; Li et al., 2023).

This *cognitive anchoring* introduces a significant bottleneck in *rare disease detection*, often leading to prolonged diagnostic delays, repeated misdiagnoses, and unnecessary interventions. Studies in clinical cognition have shown that medical decision-making is often driven by fast, heuristic-based thinking that prioritizes pattern recognition over analytical reassessment (Norman et al., 2024). This is especially problematic in the context of rare diseases, where diagnostic presentations often overlap with more common syndromes, creating fertile ground for premature closure. While previous machine learning efforts have primarily focused on enhancing accuracy through larger datasets or more powerful models (Juba & Le, 2019; Sun et al., 2017; Moreno-Barea et al., 2020), few have addressed the cognitive constraints that shape clinicians' interactions with model predictions, particularly under uncertainty. Moreover, existing studies indicate that clinicians may be unable to effectively integrate the AI's reasoning due to its opaque recommendations (Jussupow et al., 2021; Lebovitz et al., 2022), potentially exacerbating misdiagnoses (Jussupow et al., 2022).

Our work addresses two barriers to rare disease diagnosis: limited data and cognitive rigidity. We introduce a diagnostic framework that not only *detects rare diseases* but also *reduces cognitive biases*, with a focus on *anchoring*, which can impede accurate decisions. Instead of merely maximizing predictive likelihood, our system acts as a cognitive aid, encouraging clinicians to consider alternative diagnostic hypotheses. Drawing from cognitive science theories of bias mitigation (Croskerry, 2002) and leveraging recent advances in generative modeling, we design a Denoising Masked AutoEncoder (DMAE) (Wu et al., 2023) to generate plausible diagnostic counterfactuals that encourage reflective reasoning.

---

*Corresponding author. [1]The Chinese University of Hong Kong, Shenzhen, China [2]City University of Hong Kong, Hong Kong SAR, China. Correspondence to: Shuang Li <lishuang@cuhk.edu.cn>.

*Proceedings of the 43rd International Conference on Machine Learning*, Seoul, South Korea. PMLR 306, 2026. Copyright 2026 by the author(s).

Our DMAE-based model is trained on annotated clinical data to jointly capture (1) the underlying distribution of diseases and symptoms and (2) typical human diagnostic tendencies. Instead of perturbing patient representations indiscriminately, the model activates latent perturbations precisely when it detects a discrepancy between medically plausible diagnoses and those preferred by human-like diagnostic behavior. In these situations, the model explores alternative latent trajectories that remain clinically coherent but lie outside the clinician's current focus. These trajectories surface diagnostically plausible yet cognitively neglected diseases and suggest follow-up tests that meaningfully stress-test the prevailing diagnostic hypothesis. For example, the system may suggest:

> *Although common conditions can account for the current symptoms, a plausible but easily overlooked rare disease is **Ehlers–Danlos syndrome**. Consider additional tests such as **genetic screening** for connective tissue disorders. If the results are **positive**, the likelihood of this diagnosis increases substantially.*

Unlike traditional AI systems that deliver static predictions, our framework promotes active cognitive engagement, helping clinicians *break habitual diagnostic patterns* and *rethink their assumptions*. By surfacing rare yet plausible conditions, it expands the diagnostic space, fosters reflective thinking, and supports more informed clinical decisions. As Buçinca et al. (2021) have demonstrated, a mechanism that guides users to actively engage in critical thinking about initial assumptions enhances decision-making quality more effectively than merely providing predictions.

In our experiments, we evaluate the system's effectiveness using seven rare disease datasets, including four public benchmarks and three private cohorts. Our method outperformed traditional machine learning (ML) classifiers in rare disease detection while preserving optimal performance on common disease diagnosis. We validate the counterfactual hypotheses by comparing the model's hypotheses with diagnoses made by human clinicians and assessments from Large Language Models (LLMs), and expert review from a top-tier hospital. The results show that our model could identify plausible but cognitively neglected conditions, which improves diagnostic precision and supports continuing education for clinicians in primary-care and community hospitals.

## 2. Related Work

**Counterfactual Explanations.** The evolution of counterfactual explanations has transitioned from optimizing feature perturbations (Wachter et al., 2017) to frameworks that prioritize human-AI collaboration and safety. Early methods focused on generating minimal feasible changes (e.g.,

DiCE (Mothilal et al., 2020)), but were criticized for ignoring user-specific constraints and real-world applicability (Verma et al., 2021). More recent work, including Lee & Chew (2023), highlights the role of counterfactuals in mitigating cognitive biases. Lee & Chew (2023) showed that exposing users to hypothetical scenarios reduces over-reliance on erroneous AI predictions, particularly among non-experts susceptible to confirmation bias. This aligns with broader findings in human-AI interaction, where explanations must balance interpretability with decision accuracy (Buçinca et al., 2021; Straitouri et al., 2024). A significant advancement in this area is the formalization of counterfactual harm, defined as the risk that explanations may degrade human judgment. Straitouri et al. (2024) introduced structural causal models with conformal risk control to bound harmful outcomes in clinical systems. Their approach integrates monotonicity assumptions (e.g., "higher biomarker values correlate with worse prognosis") to ensure explanations align with domain knowledge, thereby addressing a gap in earlier optimization-based methods (Van Looveren & Klaise, 2021). This shift reflects a growing emphasis on safety-critical metrics, moving beyond traditional criteria like sparsity and realism (Verma et al., 2021). However, these studies mainly examine counterfactuals as post-hoc explanations or reliance interventions, whereas our work uses counterfactual generation to explicitly probe clinician anchoring in rare-disease diagnosis.

**Counterfactual Generative Models.** Generative models have been introduced to generate numerical counterfactuals, enabling dynamic adaptation to user constraints. Early GAN-based approaches, such as CounterRGAN (Nemirovsky et al., 2022), enforced immutable features via residual networks but lacked flexibility for real-time customization. FCEGAN (Hellemans et al., 2025) addresses this limitation by incorporating user-defined templates and dual discriminator losses for personalized counterfactual generation across tabular decision domains. These frameworks align with CTGAN's training-by-sampling strategy (Xu et al., 2019) to handle class imbalance, a persistent challenge in financial and medical datasets. While REVISE (Joshi et al., 2019) introduced a method for generating numerical counterfactuals using arbitrary generative models, it can produce unrealistic counterfactuals, making them unsuitable for healthcare applications, and is limited by the need for multiple calls to an optimization module. Although CFVAE (Nagesh et al., 2023) was designed for generating counterfactuals in healthcare settings using variational autoencoders, it does not account for realistic challenges in healthcare, such as class imbalance in rare disease cases and missing values in datasets. In contrast, our framework jointly models the disease distribution and a cognitive simulation of clinician decisions, and triggers counterfactual generation when plausible rare-disease hypotheses diverge

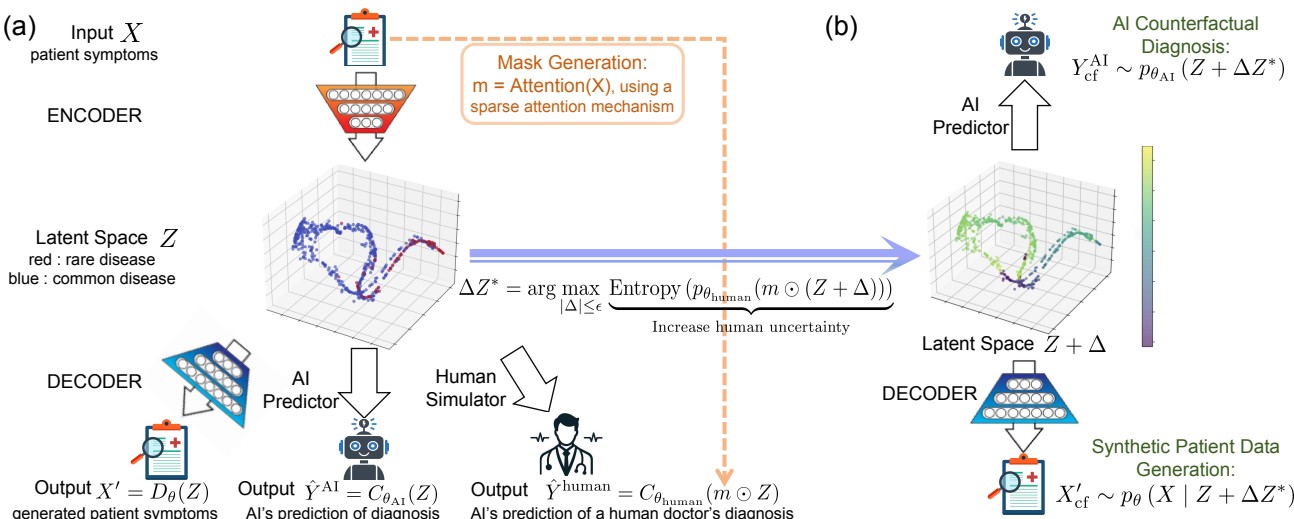

*Figure 1.* DMAE guided counterfactual reasoning framework. (a) The DMAE encodes patient features into a latent space and supports two predictors, one for the AI diagnosis and one that simulates the clinician diagnosis. (b) The method generates counterfactuals by perturbing latent codes to increase uncertainty in the clinician model. The AI predictor then outputs a counterfactual diagnosis, and the decoder reconstructs the corresponding patient profile.

from the clinician's focus, while remaining robust to the missing values and class imbalance endemic to rare-disease records.

## 3. Inherent Challenges in Modeling Rare Disease Diagnosis

In clinical diagnosis, the fundamental task is to infer the underlying disease label $Y \in \mathcal{Y}$ from observed clinical evidence $X \in \mathcal{X}$, such as patient-reported symptoms. Both human clinicians and ML models aim to learn or approximate the mapping:

$$h : X \mapsto \hat{Y}, \quad \text{where } \hat{Y} \approx \arg \max_Y P(Y \mid X).$$

By Bayes' theorem, this conditional probability can be expressed as:

$$P(Y \mid X) = \frac{P(X \mid Y) \cdot P(Y)}{P(X)},$$

where $P(Y)$ encodes prior knowledge of disease prevalence and $P(X \mid Y)$ reflects the data-generating process (e.g., symptom presentation) conditioned on a specific disease. However, in the context of *rare disease diagnosis*, this inferential process remains challenging. No matter for logistic regression, support vector machines, or deep classifiers, existing approaches face the same three critical limitations:

1. **Skewed priors.** Rare diseases typically have extremely small $P(Y)$. This prior imbalance biases both clinicians and ML models to favor common diagnoses, even when rare diseases are more plausible explanations.

2. **Overlapping symptom profiles.** Many hallmark symptoms of rare diseases (e.g., fatigue, muscle pain, or inflammation) are nonspecific and widely shared across common conditions. As a result, the likelihoods $P(X \mid Y_{\text{rare}})$ and $P(X \mid Y_{\text{common}})$ often overlap significantly, making discrimination between them highly uncertain.

3. **Incomplete evidence.** Key diagnostic features, such as genetic markers or specialized imaging, often remain absent from the record due to cost, lack of access, or simply being overlooked. This leads to an underspecified $X$, causing both humans and machines to rely on incomplete or biased feature sets. Such gaps often *reinforce* cognitive heuristics like *anchoring*, where initial impressions dominate the diagnostic path.

These challenges create a shared *algorithmic and cognitive bottleneck* for both humans and machines. Standard discriminative models $h : X \mapsto Y$, trained to directly map observed features to labels, inherit the same structural vulnerabilities as their human counterparts. Without mechanisms to uncover latent structures, handle missing information, or actively debias the inference process, both fall short in the critical task of detecting rare and underrepresented diseases.

### 3.1. Motivation for a Latent-State Generative Model

These insights motivate the need for a new kind of AI-aided diagnostic framework that can:

- *Explicitly identify cases where the observed $X$ lies in an ambiguous or overlapping region* of the feature space;

- *Hypothesize possible latent rare disease explanations* even when current evidence is incomplete;

- *Recommend additional complementary tests* (e.g., genetic panels, imaging) that can disambiguate competing diagnoses and help clinicians *break out of anchored diagnostic pathways*.

A discriminative model alone cannot meet these goals, as it is designed only to map observed input $X$ to a label prediction $\hat{Y}$ and lacks any mechanism for reasoning about uncertainty, missing data, or counterfactual information acquisition. To address these limitations, we propose a *latent-state generative model* based on the DMAE framework. This model explicitly learns a latent representation $Z$ of the patient's symptom input $X$ and generates possible reconstructions and diagnostic outcomes in a controlled, interpretable manner. This design assists both machine and human diagnostic reasoning by producing alternative hypotheses, with emphasis on rare conditions that can be missed because of low prior probability or heuristic bias.

The proposed latent-state generative model takes the following form (as illustrated in Figure 1):

- **Input:** $X$ (observed patient symptoms)

- **Latent state:** $Z$ (learned stochastic representation of patient condition)

- **Outputs:**

  1. $X'$: A reconstructed or generated version of patient symptoms (counterfactual or prototypical symptom set)
  2. $\hat{Y}^{\mathrm{AI}}$: Prediction of the true diagnosis based on latent state $Z$
  3. $\hat{Y}^{\mathrm{human}}$: Model's simulation of a human doctor's likely diagnostic decision

## 4. Our Proposed Generative Model Formulation

We assume access to a dataset of triplets $\left\{\left(X_i, Y_i^{\mathrm{human}}, Y_i^{\mathrm{true}}\right)\right\}_{i=1}^N$, where $X_i \in \mathbb{R}^d$ represents patient features, $Y_i^{\mathrm{true}} \in \{1, \ldots, C\}$ is the ground-truth diagnosis, and $Y_i^{\mathrm{human}}$ denotes the clinician's early, potentially anchored diagnosis formulated under conditions of incomplete information, modeled as a preliminary, cue-limited judgment state rather than an ideal final expert label. Our goal is to learn a latent state generative model with three key components: the patient's latent diagnostic state $Z$, the clinician's decision $Y^{\mathrm{human}}$, and the AI's prediction $Y^{\mathrm{AI}}$. By explicitly modeling the cognitive gap between human and AI reasoning, the model enables discrepancy-aware inference and supports bias-aware diagnostic support.

$$p_\theta\left(X, Y^{\mathrm{AI}}, Y^{\mathrm{human}}, Z\right) = p(Z)\, p_\theta(X \mid Z)\, p_\theta\left(Y^{\mathrm{AI}} \mid Z\right)$$
$$\cdot\, p_\theta\left(Y^{\mathrm{human}} \mid \tilde{Z}\right).$$

Following standard latent-variable generative modeling, this factorization is adopted for tractability rather than asserting real-world clinical independence. Here, $Z \in \mathbb{R}^k$ is a latent representation inferred from $X$, and $\tilde{Z}$ denotes a modulated version of $Z$. Although humans and AI observe the same input $X$, their predictions can diverge due to: (1) cognitive load limiting human attention to parts of $X$, and (2) fundamentally different mapping functions. We explicitly reflect these factors in the design of our DMAE-based generative model.

**Latent Representation Learning with Denoising Masked AutoEncoder**  Given that real-world clinical inputs $X \in \mathbb{R}^d$ often contain missing or underreported features, particularly for rare diseases, we employ DMAE (Wu et al., 2023) to learn a robust and informative latent representation $Z$.

For each observed input $X_i$, we sample a binary mask $r_i \in \{0,1\}^d$ to randomly drop a subset of observed entries, simulating incomplete or noisy records. The resulting corrupted input is $\tilde{X}_i = r_i \odot X_i$, which is then encoded to a latent distribution $q_\phi\left(Z_i \mid \tilde{X}_i\right)$. The decoder reconstructs the full input, and the reconstruction loss is computed only on the originally observed (i.e., uncorrupted) entries:

$$\mathcal{L}_{\mathrm{recon}} = \mathbb{E}_{q_\phi\left(Z_i \mid \tilde{X}_i\right)}\left[\left\|(1 - r_i) \odot \left(X_i - \hat{X}_i\right)\right\|_2^2\right]$$

This approach helps the model infer missing or overlooked features, similar to masked token prediction in language models, and it learns robust task-relevant representations. The resulting embeddings generalize across settings and support downstream tasks such as diagnosis prediction and modeling human-AI divergence.

**Dual Classification Losses**  The latent code $Z_i$ is leveraged to predict two diagnostic outcomes: the *ground-truth diagnosis* $Y_i^{\mathrm{true}}$, and the *observed human diagnosis* $Y_i^{\mathrm{human}}$. We define two separate classification objectives:

- **AI Prediction Loss (truth-matching):**
$$\mathcal{L}_{\mathrm{AI}} = -\mathbb{E}_{q_\phi(Z_i \mid X_i)}\left[\sum_c \alpha_c (1 - p_c)^\gamma \log p_c\right],$$
$$\alpha_c \propto \frac{1}{\mathrm{freq}(c)}.$$

Here, $p_c = p_{\theta_{\mathrm{AI}}}\left(Y_i^{\mathrm{true}} = c \mid Z_i\right)$ denotes the predicted probability of class $c$ under the AI classifier. This objective encourages the model to leverage the *full latent representation $Z_i$* to generate accurate, clinically grounded predictions aligned with the ground-truth diagnosis, using a classifier parameterized by $\theta_{\mathrm{AI}}$.

To address class imbalance, which is severe in rare disease settings, we adopt a focal loss variant (Lin et al., 2017) that down-weights easy examples and focuses learning on hard or ambiguous cases. We set $\alpha_c$ proportional to the inverse empirical class frequency, where $\text{freq}(c)$ denotes the prevalence of disease $c$ in the training set (i.e., the fraction of samples with $Y^{\text{true}} = c$). This calibrated objective increases sensitivity to underrepresented classes and supports more exploratory inference, which helps the model surface atypical or underrecognized diagnostic patterns that may otherwise be missed. In this way, the AI component functions not only as a predictor but also as a hypothesis generation aid for clinical decision making.

- **Human Simulation Loss (cognitive-matching):**

$$\mathcal{L}_{\text{human}} = \mathbb{E}_{q_\phi(Z_i|X_i)} \left[ -\log p_{\theta_{\text{human}}} \left( Y_i^{\text{human}} \mid \tilde{Z}_i \right) \right]$$

Here, $\tilde{Z}_i = m_i \odot Z_i$ is a selectively masked version of the latent vector, where the learned attention mask $m_i \in [0,1]^k$ gates which latent dimensions are used by the human prediction head. This reflects the idea that, given the same input $X_i$, *humans and AI may focus on different parts of the data and apply distinct cognitive functions to reach a diagnosis*.

Importantly, the prediction functions for AI and human simulation are parameterized separately, using $\theta_{\text{AI}}$ and $\theta_{\text{human}}$ respectively. This architectural asymmetry captures both attentional differences (via $m_i$) and functional differences in diagnostic reasoning, allowing us to explicitly model and analyze human-AI cognitive divergence.

**Modeling Human-AI Cognitive Gaps via Sparse Self-Attention Mask**  Specifically, we compute the attention mask $m_i$ using a learnable self-attention module:

$$u_i = \text{Softmax}\left( \frac{Q\left(X_i\right) K\left(X_i\right)^\top}{\sqrt{d}} \right) V\left(X_i\right)$$

$$m_i = \sigma(W_m u_i + b_m)$$

where $Q(\cdot), K(\cdot), V(\cdot)$ are linear projections (as proposed in Vaswani et al. (2017)) that produce query, key, and value vectors from the input $X_i$, and the output $u_i$ is pooled to form a $k$-dimensional attention vector, which is then mapped to $m_i \in (0,1)^k$ via a sigmoid gate. This attention mechanism identifies which latent features humans are likely to focus on, given the current case. This mask is an interpretable approximation of selective cue utilization, not a direct measurement of true human attention.

To ensure interpretability and mimic human cognitive constraints, we impose an $\ell_1$ sparsity penalty on the attention mask:

$$\mathcal{L}_{\text{mask}} = \lambda_{\text{mask}} \cdot \|m_i\|_1$$

This encourages the human prediction head to rely on a small subset of salient features, reflecting *limited cognitive bandwidth* and enhancing the *interpretability* of human diagnostic pathways.

**Contrastive Learning for Rare Disease Separability**  To prevent rare disease embeddings from collapsing into common clusters, we introduce a contrastive loss:

$$\mathcal{L}_{\text{contrast}} = \sum_{(i,j,k)} \max\left(0,\ \delta + d(Z_i, Z_j) - d(Z_i, Z_k)\right),$$

where $Z_i$ and $Z_j$ are latent representations from the same rare disease class, and $Z_k$ is from a common disease class.

This loss encourages embeddings of the same rare class to remain close while pushing them away from embeddings of common classes, thereby promoting greater separability and preserving the distinctiveness of rare conditions in the latent space.

**Cognitive Gap Identification: Discrepancy Between AI and Human Attention**  To quantify the discrepancy between AI and clinician reasoning, particularly in rare disease cases, we introduce a *cognitive gap loss*. This loss encourages the AI model to attend to features that may be under-utilized by human clinicians, highlighting potential diagnostic blind spots. Formally, we define the loss as:

$$\mathcal{L}_{\text{gap}} = \sum_{i:\, Y_i^{\text{true}} \in \text{rare}} \left\| m_i \odot \nabla_{Z_i} \log p_{\theta_{\text{AI}}}(Y_i^{\text{true}} \mid Z_i) \right\|_2^2,$$

where $Z_i$ is the latent representation, $m_i \in [0,1]^k$ is the learned attention mask approximating human focus, and $\nabla_{Z_i} \log p_{\theta_{\text{AI}}}(Y_i^{\text{true}} \mid Z_i)$ captures the sensitivity of the AI's prediction to each latent feature.

By penalizing high-gradient regions aligned with human attention $m_i$, the model is encouraged to focus on dimensions that are often overlooked, especially in the context of rare diseases. This fosters attentional divergence in rare disease cases, where the AI can uncover atypical patterns that clinicians might miss due to cognitive biases.

## 4.1. Total Objective and Training Curriculum

The overall loss function is defined as:

$$\mathcal{L}_{\text{total}} = \mathcal{L}_{\text{rec}} + \mathcal{L}_{\text{AI}} + \mathcal{L}_{\text{human}} + \gamma \mathcal{L}_{\text{contrast}} + \eta \mathcal{L}_{\text{mask}} + \xi \mathcal{L}_{\text{gap}}.$$

The training process follows a staged curriculum: starting with the DMAE warm-up using reconstruction loss, then introducing focal loss for rare disease prediction. The curriculum then adds human cognitive modeling and sparsity regularization, followed by contrastive learning for separating rare and common diseases. Finally, the cognitive

gap loss is incorporated to address attention mismatches between AI and human clinicians.

We will train the DMAE using the above loss function. Given the learned generative DMAE model, we can design the following counterfactual generation tasks.

## 5. Counterfactual Generation for Cognitive Anchoring Correction

To mitigate diagnostic errors from cognitive anchoring, we introduce a counterfactual generation mechanism that leverages the model's probabilistic structure. Given patient data $X$, if $p_{\theta_{\mathrm{AI}}}$ assigns relatively high probability to a plausible diagnosis $Y_{\mathrm{AI}}$, especially a rare or under-considered alternative that differs from the clinician's current diagnosis, this triggers counterfactual generation to challenge the initial decision of human and guide follow-up evaluation or testing.

The **goal** of the counterfactual generation here is to

*Disrupt doctors' fixation on initial hypotheses by generating alternative diagnostic pathways, particularly for rare diseases.*

**Learning Optimal Perturbation**  The perturbation is learned to increase uncertainty in the human (or human-approximating) model, thus exposing cognitive blind spots.

$$\Delta Z^* = \arg \max_{\|\Delta\| \leq \epsilon} \underbrace{\mathrm{Entropy}\left(p_{\theta_{\mathrm{human}}}(m \odot (Z + \Delta))\right)}_{\text{Increase human uncertainty}}$$

Here, $\|\Delta\| \leq \epsilon$ bounds the locality of the perturbation in latent space, ensuring the changes remain within a medically interpretable range, while physiological validity is additionally enforced in the decoded feature space via differentiable penalties on out-of-range reconstructed variables; only counterfactuals passing both checks are retained (details in Appendix C.4). Without perturbation, the AI's prediction from the original $Z$ may align closely with the clinician's current belief. By contrast, perturbing $Z$ explores latent variations that introduce diagnostic ambiguity from the human's perspective-potentially uncovering under-recognized or rare conditions.

**Counterfactual Output Generation**  Once the optimal perturbation $\Delta Z^*$ is obtained, the system generates two outputs:

- **AI Counterfactual Diagnosis**

$$Y_{\mathrm{cf}}^{\mathrm{AI}} \sim p_{\theta_{\mathrm{AI}}}(Z + \Delta Z^*)$$

This may yield a rare disease prediction that prompts reconsideration of the original diagnosis.

- **Synthetic Patient Data Generation** A DMAE is used to reconstruct the corresponding patient profile:

$$X_{\mathrm{cf}}' \sim p_{\theta}\left(X \mid Z + \Delta Z^*\right)$$

Here, $X_{\mathrm{cf}}'$ represents a plausible synthetic patient who presents similarly but includes key missing symptoms supporting the rare disease.

Finally, the system communicates the counterfactual insight as:

*"Consider alternative diagnoses with similar presentations: [AI-suggested disease $Y_{\mathrm{cf}}^{\mathrm{AI}}$]. If additional findings such as $X_{\mathrm{cf}}'$ were observed, the likelihood of this condition would increase to $p_{\theta_{\mathrm{AI}}}(Y_{\mathrm{cf}}^{\mathrm{AI}} \mid Z + \Delta)$."*

This form of explanation aims to encourage the clinician to reflect, reassess, and refine their diagnostic reasoning with evidence-informed support from the AI.

## 6. Experiment

To evaluate our framework, we conduct extensive experiments to (i) assess whether counterfactual explanations help address cognitive gaps and support clinical decision-making, and (ii) validate robust diagnostic performance for *rare disease* detection. Model architecture and training details are provided in Appendix B and Appendix C.

We evaluate our framework across seven rare disease datasets. This collection comprises four public rare disease benchmarks: (i) **granulomatosis with polyangiitis (GPA)** from the Hannover Medical School dataset (Chen et al., 2024), (ii) **IHPRF3** derived from literature-based case reports (Robinson et al., 2020), (iii) **cerebro-costo-mandibular syndrome (CCMS)** from the Matchmaker Exchange federated network (Philippakis et al., 2015; Buske et al., 2015), and (iv) **multiple myeloma** extracted from the MIMIC-IV-Rare cohort (Zhao et al., 2026). Furthermore, we expand our evaluation to three private, real-world cohorts constructed in a top-tier hospital, covering **Gitelman syndrome**, **acromegaly**, and **hypertrophic cardiomyopathy (HCM)**. Detailed descriptions of each dataset are provided in Appendix A.

### 6.1. Counterfactual Intervention for Cognitive Anchoring Correction

#### 6.1.1. Clinical Use Cases Enabled by Counterfactual Support

Rare disease diagnosis often involves missing evidence, clinician–model disagreement, and high uncertainty. Our framework enables counterfactual analysis in three practical use cases, which we use for evaluation.

**Use Case 1: Evidence Completion Under Low Confidence.** When key indicators are missing, both clinicians and models can produce low-confidence predictions and default to common diagnoses. Our method generates counterfactuals that simulate the presence of targeted follow-up tests (e.g., genetic panels or imaging proxies) and quantifies the resulting change in diagnostic probabilities. This analysis highlights which additional evidence is most likely to disambiguate common and rare conditions.

**Use Case 2: Resolving AI–Clinician Disagreements.** When the AI prediction differs from the clinician diagnosis, we generate counterfactual profiles that shift the human simulation toward the AI alternative while remaining within a bounded perturbation. The resulting contrastive examples expose the latent dimensions and reconstructed features that drive the discrepancy, providing interpretable evidence for reconciliation.

**Use Case 3: Uncertainty Guided Alternative Hypotheses.** For cases in which the clinician model exhibits high uncertainty, we perturb the latent code in directions that maximally increase uncertainty and sample multiple plausible counterfactual profiles. The resulting set provides a structured set of alternative hypotheses and associated evidence patterns, which encourages broader differential consideration and reduces reliance on anchored reasoning.

### 6.1.2. QUALITATIVE ASSESSMENT

In this section, we assess DMAE-guided counterfactual reasoning across all datasets. To obtain a comprehensive evaluation of the generated counterfactuals, we assess them with both LLM-based reviewers and clinical experts. We summarize representative cases spanning the three use cases in Figure 2, with the complete set of physician-reviewed examples provided in Appendix H. Board-certified specialists reviewed 30 counterfactual cases across the seven cohorts, of which approximately 90% were rated clinically plausible and useful; the remaining cases were judged of low marginal utility for senior experts rather than clinically implausible. Details on LLM prompts and full model responses appear in Appendix D.

Across cohorts, the model adjusts low confidence predictions by simulating the effect of missing evidence and elevating plausible rare diagnoses, which helps counteract cognitive anchoring. In disagreement cases, the generated flipped counterfactuals isolate the smallest changes that shift the clinician model toward the AI alternative, which clarifies the features that drive the mismatch. In high-uncertainty cases, the method generates multiple plausible profiles that correspond to distinct diagnostic outcomes, which supports differential consideration and reduces reliance on a single anchored hypothesis.

**LLM-Based Evaluation** We use an LLM as an automated reviewer to qualitatively assess the generated counterfactuals (Gao et al., 2025). Specifically, we use GPT-4o with temperature set to 0.1 to reduce stochasticity. Using structured prompts, the LLM provides brief, criterion-guided justifications that assess clinical plausibility, case relevance, and whether the counterfactual encourages de-anchored diagnostic reasoning and actionable follow-up.

**Evaluation by Doctors** Clinical experts from a leading hospital review the generated counterfactuals for medical plausibility and case relevance. Two board-certified specialists from the relevant department conduct a consensus-based assessment and resolve disagreements through joint discussion. The review follows a single-blind protocol. Reviewers receive only the patient profile and the generated counterfactuals, without access to model details or study hypotheses.

**Latent Space Analysis** To better understand the mechanisms behind counterfactual generation, we analyze the

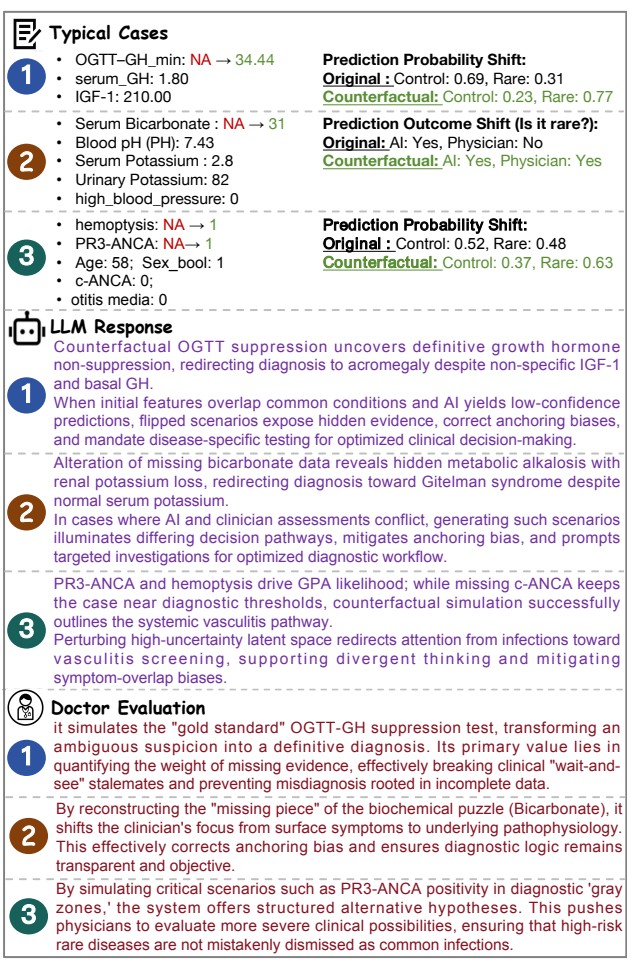

*Figure 2.* Example LLM rationales and clinician evaluations for three use cases.

learned latent space (Appendix E for the full model and Appendix F for ablations). Effective counterfactual generation requires a structured latent geometry that links clinical observations to diagnostic concepts.

### 6.1.3. QUANTITATIVE ASSESSMENT

**Baselines** We compare our method with two counterfactual generation baselines. REVISE (Joshi et al., 2019) performs gradient-based optimization in the latent space of a generative model, and CFVAE (Nagesh et al., 2023) jointly trains a variational autoencoder with a binary predictor to produce counterfactuals.

**Metrics** We evaluate counterfactual quality using two metrics. (1) **Label Flip Rate**: the fraction of generated counterfactuals that the target classifier assigns to the intended target class, which measures validity. (2) **RMSE**: the root mean squared error between the counterfactual and the original input, which measures perturbation size, where lower values indicate higher proximity. For binary or multi-hot categorical cohorts, RMSE is used mainly as a benchmark-consistent proxy for perturbation magnitude rather than a complete measure of semantic or clinical similarity, so we interpret it together with label-flip validity and expert-reviewed qualitative examples.

**Results** Table 1 reports results for our method, REVISE, CFVAE, and an ablation across seven datasets. Our method achieves the highest label flip rate and the lowest RMSE, which indicates that it produces more valid counterfactuals with smaller perturbations than the baselines. For a qualitative side-by-side comparison, we additionally provide nine matched label-flipping examples generated by REVISE, CFVAE, and our framework on the same patient inputs in Appendix I.

### 6.2. Robust Rare Disease Detection Under Extreme Imbalance

**Datasets and baselines.** Rare disease prediction is severely class-imbalanced, which challenges standard classifiers. To test robustness under extreme imbalance, we report results on the two most imbalanced datasets in our study: Gitelman and HMS. For Gitelman, the rare-to-common ratio ranges from 94:100 to 94:500. For HMS, we detect GPA against non-GPA controls from the Hannover Medical School cohort, with ratios from 11:12 to 11:82. All baselines use standard imbalance mitigation: focal loss for neural networks, SMOTE for SVM and logistic regression, and class-weighting for XGBoost and LightGBM.

**Results** Our approach consistently outperforms the other classifiers on both cohorts, as shown in Figure 3. We report AUC and PRAUC. Notably, performance remains stable,

*Table 1.* Quantitative performance metrics across seven rare diseases.

| | GPA | | IHPRF3 | |
|---|---|---|---|---|
| Model | Flip ↑ | RMSE ↓ | Flip ↑ | RMSE ↓ |
| REVISE | 0.94±0.07 | 0.18±0.06 | 0.99±0.00 | 0.14±0.04 |
| CFVAE | 0.85±0.18 | 0.29±0.09 | 0.95±0.05 | 0.64±0.01 |
| **Ours** | **1.00±0.00** | **0.12±0.03** | **1.00±0.00** | **0.10±0.04** |
| Ablation | **1.00±0.00** | 0.25±0.07 | **1.00±0.00** | 0.47±0.12 |
| | CCMS | | Multiple Myeloma | |
| Model | Flip ↑ | RMSE ↓ | Flip ↑ | RMSE ↓ |
| REVISE | 1.00±0.00 | 0.26±0.07 | 0.98±0.01 | 0.31±0.02 |
| CFVAE | 0.98±0.01 | 0.79±0.01 | 0.95±0.02 | 0.38±0.01 |
| **Ours** | **1.00±0.00** | **0.14±0.08** | **1.00±0.00** | **0.10±0.09** |
| Ablation | **1.00±0.00** | 0.27±0.06 | **1.00±0.00** | 0.39±0.07 |

| | Gitelman | | Acromegaly | | HCM | |
|---|---|---|---|---|---|---|
| Model | Flip ↑ | RMSE ↓ | Flip ↑ | RMSE ↓ | Flip ↑ | RMSE ↓ |
| REVISE | 0.96±0.03 | 5.40±0.89 | 0.92±0.11 | 13.96±14.4 | 0.70±0.40 | 0.33±0.04 |
| CFVAE | 0.96±0.02 | 12.0±1.77 | 0.85±0.15 | 13.96±14.8 | 0.80±0.40 | 0.33±0.01 |
| **Ours** | **1.00±0.00** | **1.93±0.76** | **1.00±0.00** | **0.18±0.10** | **1.00±0.00** | **0.10±0.13** |
| Ablation | **1.00±0.00** | 4.85±3.27 | **1.00±0.00** | 0.21±0.08 | **1.00±0.00** | 0.46±0.27 |

and in some settings improves, as the degree of imbalance increases. This trend suggests that additional common disease samples sharpen decision boundaries in the latent space, which improves the separation of rare disease signatures and can turn data skew into a learning advantage.

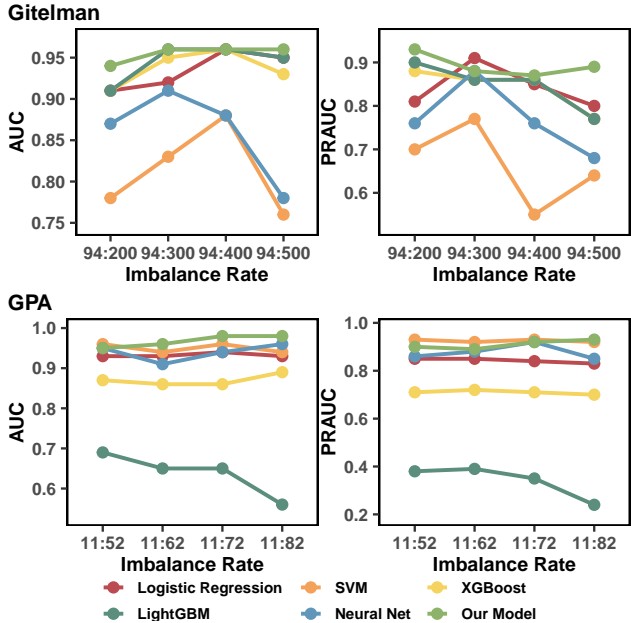

*Figure 3.* Comparison of model performance under varying imbalance ratios for Gitelman (private) and HMS (public) datasets.

**Ablation Study** To assess the contribution of each component, we perform ablations that remove individual loss terms during fine-tuning. We report AI and human predictor performance for all ablations across seven datasets in Appendix G. The full model achieves consistently strong results across cohorts. Removing AI-specific objectives

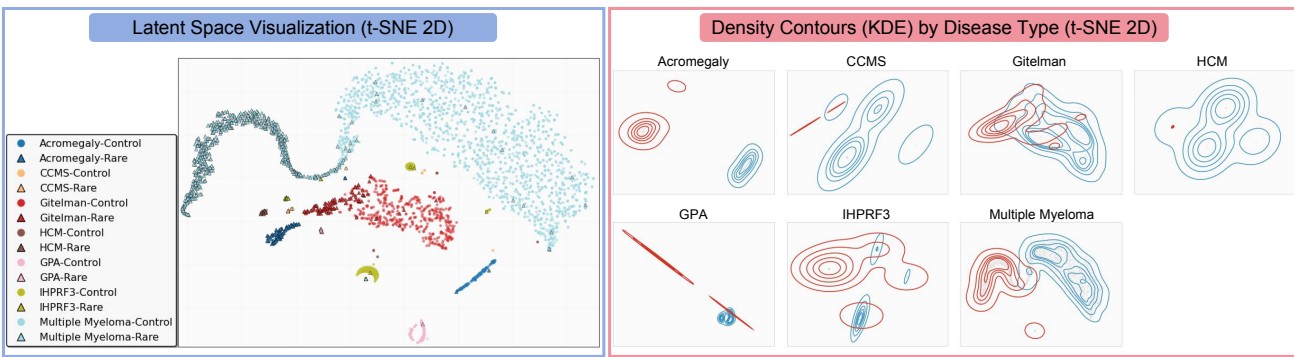

*Figure 4.* Unified DMAE latent space under pooled training. Left: 2D t-SNE projection colored by disease category and case-control status, showing disease-level clustering. Right: KDE contours for each disease, showing separation between rare cases (red) and controls (blue) across pooled cohorts. These visualizations illustrate latent structure rather than perfect class separation; quantitative results provide the primary evidence of model behavior.

substantially degrades the AI predictor, whereas removing human-centric objectives or mask regularization degrades the human simulator.

### 6.3. Pooled Training for a Shared Latent Space Across Disease Domains

To evaluate the robustness and scalability of our DMAE framework beyond task-specific boundaries, we conducted a "Foundation-style" experiment by pooling all seven datasets. This experiment assesses whether a unified model can learn a shared latent space across highly heterogeneous feature sets and hundreds of disease categories.

**Unified input construction.** Each cohort contains a different set of clinical variables, so we form a single feature set by taking the union across all datasets. For each patient, we create a fixed-length input over this shared feature set and include a binary missingness indicator for every entry. Missing variables use a placeholder value, while the indicator specifies whether the value is observed.

**Latent representation analysis.** We visualize the shared latent space with a 2D t-SNE projection and use KDE contours to compare case and control distributions for each disease. Figure 4 (left) shows clustering by disease category, which suggests the encoder captures disease-specific signatures despite heterogeneous inputs. Figure 4 (right) shows KDE contours separate rare cases (red) from controls (blue) for each condition, indicating pooled training preserves disease level separability even when cohorts overlap.

**Predictive performance on pooled data.** Table 2 reports mean PRAUC across cohorts. The unified model remains competitive with task-specific models. We also compare the Unified DMAE with discriminative baselines trained on the same pooled data, including XGBoost, SVM, and an MLP (Detailed visual comparisons are provided in Appendix J.1). While these baselines degrade under the sparsity and di-

mensionality of the pooled feature space, our generative framework achieves higher AUC and stronger recall for rare conditions by sharing information across domains.

*Table 2.* Comparison of Task-Specific vs. Unified DMAE (Metric: Mean PRAUC).

| Dataset Category | Task-Specific | Unified DMAE | Relative Change |
|---|---|---|---|
| Public Datasets | 0.93 | 0.90 | -3.23% |
| Private Datasets | 0.96 | 0.95 | -1.04% |

**Anchoring Targeted Counterfactuals in the Unified Setting.** We also evaluate counterfactual reasoning under pooled training. For each patient, we choose the counterfactual target as the rare disease class with the highest predicted probability among candidate classes. As detailed in Appendix J.2, the Unified DMAE continues to generate clinically plausible counterfactuals even after pooling seven heterogeneous datasets. The resulting examples remain consistent with the behavior of task-specific models and help expose diagnostic blind spots by showing how missing or atypical evidence can shift the predicted diagnosis away from an anchored initial hypothesis.

## 7. Conclusion

We introduced a cognitive-aware counterfactual reasoning framework that perturbs latent patient representations via a DMAE-based latent state generative model to counter cognitive anchoring in rare disease diagnosis. By generating realistic "what-if" scenarios, our method surfaces overlooked conditions and guides clinicians toward alternative hypotheses. A mixed LLM- and doctor-based evaluation confirms the scientific soundness and clinical relevance of the generated cases. This framework fosters reflective diagnostic reasoning, enhances interpretability, and offers a scalable tool for bridging human knowledge gaps in challenging medical scenarios.

## Acknowledgments

This work was supported in part by the Key Program of the National Natural Science Foundation of China (NSFC) under Grant No. 72495131; the Shenzhen Stability Science Program 2023, Shenzhen Key Lab of Multi-Modal Cognitive Computing; the Shenzhen Science and Technology Program No. JCYJ20250604141038013; and the Longgang District Key Laboratory of Intelligent Digital Economy Security.

## Impact Statement

This study aims to address the underdiagnosis of rare diseases caused by cognitive biases in clinical decision-making. Our framework helps clinicians consider rare conditions more effectively through generative counterfactuals, potentially reducing diagnostic delays and improving patient outcomes, especially in underserved areas with limited specialized expertise. By modeling the cognitive gaps between humans and AI, it promotes transparent and bias-aware collaboration, setting a practical example for AI applications in healthcare and other high-stakes fields.

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

# A. Experimental Datasets

To evaluate our method, we consider the following seven datasets.

**Gitelman Syndrome** This dataset comprises real clinical records from a top hospital, focusing on Gitelman syndrome (GS), a rare autosomal recessive renal tubulopathy. The data contains 594 patients, including 94 diagnosed with GS and 500 non-GS individuals. Five key diagnostic features are included: *Serum Potassium*, *Urine Potassium*, *pH*, *Bicarbonate*, and *High Blood Pressure*, with labels derived from clinical diagnoses. To emulate real-world scenarios where critical test results are missing (a common challenge in rare disease diagnosis), we retain the missing values in the original data. This enables counterfactual analysis to quantify how missing tests impact predictions, thereby guiding clinicians to prioritize specific examinations for undiagnosed cases. The dataset is split into 80%-20% train-test sets for GS classification, with subsequent counterfactual perturbation analysis performed in the latent space of the complete data. It should be noted that we retained the situation of data imbalance, which is consistent with the situation that the incidence of rare diseases in the real world is much lower. And despite this imbalance, our model still maintained good performance.

**Acromegaly** This dataset includes real-world clinical records from a top hospital, focusing on acromegaly, a chronic disorder caused by excessive growth hormone (GH) secretion, typically due to pituitary somatotroph adenomas. The data contains 181 patients, comprising 88 diagnosed with acromegaly and 93 non-acromegaly controls. Three clinically significant features are incorporated: *Serum GH*, *IGH-1*, and *OGTT-GH_min*, with labels derived from clinical diagnoses. To reflect realistic data incompleteness, we retain naturally occurring missing values in the original dataset and explicitly record their positions. This facilitates counterfactual generation that aligns with clinical practice, allowing clinicians to evaluate how incomplete laboratory profiles influence diagnostic predictions. The dataset is partitioned into 80%-20% training-test sets for binary classification, followed by counterfactual perturbation and interpretability analysis in the latent space of the complete data to identify critical diagnostic drivers.

**Hypertrophic Cardiomyopathy (HCM)** This dataset includes real-world clinical records from a top hospital, focusing on hypertrophic cardiomyopathy (HCM), an inherited cardiac disorder characterized by abnormal myocardial thickening that may lead to ventricular outflow tract obstruction, arrhythmias, and heart failure. The data contains 36 patients, including 21 HCM-diagnosed individuals and 15 individuals with another rare disease (ATTR, amyloidosis trans-thyretin related) as the control group. Eight clinically significant features are incorporated: *Asymmetric Hypertrophy*, *SAM*, *Low Left Ventricular Voltage*, *High Left Ventricular Voltage*, *Family History*, *Sarcomere Gene Mutation*, *TTR Gene Mutation*, and *Amyloid Deposition*. Similarly, to preserve clinical authenticity, naturally occurring missing values in the original dataset are retained and explicitly mapped for interpretability. The dataset is partitioned into 80%-20% training-test splits for HCM classification. Post-training, counterfactual perturbation and causal analysis are conducted in the latent space of the complete data to identify critical diagnostic patterns and feature interactions.

**Hannover Medical School (HMS)** This dataset is derived from Chen et al. (2024) and contains real-world clinical records targeting granulomatosis with polyangiitis (GPA), which is an ANCA-associated vasculitis frequently linked to PR3-ANCA and upper-airway/pulmonary involvement. The cohort includes 93 subjects in total, comprising 11 GPA patients and 82 non-GPA controls selected for their high clinical confusability with GPA in ENT and respiratory presentations. To better capture the informative nature of clinical testing, we have refined the representation of key clinical indicators, specifically *Hemoptysis*, *Proteinase 3 Antibody Titer (PR3-ANCA)*, and *Cytoplasmic ANCA (c-ANCA)*, by transitioning from simple binary encoding to a dual-column tri-state format (*measured* and *abnormal*). This structure explicitly distinguishes among three clinical states: (i) not measured, (ii) measured with normal results, and (iii) measured with abnormal findings, thereby eliminating the inherent ambiguity between missing values and negative observations. Other clinically meaningful features, including *Otitis Media*, *Knee Pain (Bilateral)*, *Peripheral Cyanosis*, and *Rhinitis*, are also incorporated. The dataset is split into an 80%–20% training–test partition for binary GPA classification. Subsequent to the classification task, counterfactual perturbation and interpretability analysis are conducted in the latent space of the completed data to identify key diagnostic drivers and interactions between symptoms and serological markers.

**LIRICAL (Literature Case Reports)** This dataset originates from the evaluation set introduced alongside LIRICAL (Robinson et al., 2020), consisting of 384 literature-derived case reports curated into structured phenotype profiles. Each case is annotated with a set of Human Phenotype Ontology (HPO) terms and a confirmed Mendelian diagnosis (OMIM), providing a standardized benchmark for phenotype-driven rare-disease identification. For our study, we construct a binary classification task targeting *Hypotonia, infantile, with psychomotor retardation and characteristic facies 3* (IHPRF3). We first convert all phenotypic terms appearing in the dataset into a unified 0/1 multi-hot representation, where each feature indicates the presence or absence of a symptom in a given case. We then refine the feature space by retaining (i) phenotype

features clinically related to IHPRF3 (e.g., *Developmental regression*, *Intellectual disability*, *Severe muscular hypotonia* and etc.) and (ii) additional frequently occurring phenotype features that capture common background clinical patterns across cases. The dataset contains 10 IHPRF3 cases as positive samples, while all remaining cases are treated as negatives (*non-IHPRF3*), and is split into an 80%–20% training–test partition for binary classification.

**Matchmaker Exchange (MME)** Matchmaker Exchange (MME) is a federated rare-disease case-sharing framework that supports cross-database matchmaking based on phenotypic and genotypic similarity, enabling the identification of clinically comparable cases across different institutions (Philippakis et al., 2015; Buske et al., 2015). In this work, we use an MME benchmark subset containing 50 records, where each record is provided with structured phenotype descriptions and an associated diagnosed disorder. For this dataset, we first extract the union of all phenotype features appearing in the dataset and convert each case into a 0/1 multi-hot representation. We then formulate a binary classification task by selecting *Cerebro-costo-mandibular syndrome* (CCMS; *117650 CEREBROCOSTOMANDIBULAR SYNDROME ;;CCM SYNDROME; CCMS;; RIB GAP DEFECTS WITH MICROGNATHIA*) as the target class. The dataset includes 12 CCMS records as positive samples, while the remaining records are treated as negatives (*non-CCMS*). To focus on clinically meaningful evidence, we retain phenotype features medically relevant to CCMS presentations, such as *Cleft palate*, *Feeding difficulties*, and *Pierre-Robin sequence*, among other related manifestations. Finally, the dataset is split into an 80%–20% training–test partition for binary CCMS classification.

**MIMIC-IV-Rare** This dataset is derived from Zhao et al. (2026) and curates de-identified MIMIC-IV clinical records into structured phenotype features for rare-disease reasoning. After minor normalization (e.g., case-sensitive string cleanup) and additional preprocessing, our processed MIMIC-IV-Rare contains 12,818 records covering 4,885 rare diseases and 15,207 phenotype features. We formulate a binary classification task for *multiple myeloma*, using 596 *multiple myeloma* records as positives. and selecting 2,000 non-*multiple myeloma* records from other diseases that exhibit symptom patterns similar to *multiple myeloma* as negative samples. To focus the model on clinically relevant evidence, we retain the 50 most informative phenotype features closely associated with *multiple myeloma* and its look-alike conditions, reflecting common comorbidities and manifestations (e.g., *Hypertension*, *Anemia*, *Fatigue*, *Fever*, *Back Pain*, and *Peripheral Neuropathy*). The dataset is partitioned into an 80%–20% training–test split for binary classification, followed by latent-space counterfactual perturbation and interpretability analysis to identify key diagnostic patterns and feature interactions.

**Acquisition of Human Diagnostic Labels** To model the discrepancy between machine and clinician reasoning, we define human labels ($Y^{human}$) as preliminary, potentially anchored diagnostic decisions formulated under incomplete information. For private clinical cohorts, $Y^{human}$ represents authentic initial clinical impressions extracted from Electronic Health Records (EHR) during a patient's first visit. For public benchmarks where initial impressions were not explicitly recorded, we generate surrogate human labels using a Logistic Regression (LR) model. We adopt LR as a proxy for heuristic-based thinking because its linear nature intentionally simulates a "cognitive anchor" by prioritizing salient, high-frequency symptom patterns over the complex latent-state reasoning required for rare disease identification. This use of a transparent, lower-capacity linear model to approximate cue-limited, anchor-prone clinician judgment is consistent with prior work on heuristic medical decision-making and linear models of physician judgment (Marewski & Gigerenzer, 2012; Wigton, 1988).

# B. Model Architecture Details

## B.1. DMAE Architectures

The Denoising Masked AutoEncoder (DMAE) architecture captures clinical feature mappings through an Encoder and Decoder. The Encoder uses ELU activations to project raw features into a 32-dimensional latent space, while the Decoder reconstructs inputs from this space. Categorical features are embedded via a dedicated layer, and the design supports robust learning from incomplete data. Take the Gitelman syndrome dataset as an example, key components are detailed in Table 3, which outlines layer dimensions and functional roles.

## B.2. Predictor Architectures

The AI and human predictors, along with the attention mask network, are designed to explicitly model the divergence between machine and clinician reasoning. The AI predictor operates in the full latent space to generate ground truth-aligned diagnoses, while the human predictor uses a sparse attention mask (generated by the mask network) to simulate cognitive constraints in clinical decision making. Table 4 outlines the architecture details, including layer dimensions, activation functions, and the attention mechanisms. This modular design supports interpretable counterfactual generation by isolating

*Table 3.* DMAE architecture configuration

| Component | Layers | Dimension | Functional Description |
|---|---|---|---|
| Encoder | Input Layer | 5 | Raw clinical features |
| | Hidden Layer | 128 | ELU-activated transformation: $h = \text{ELU}(Wx + b)$ |
| | Latent Space | 32 | Bottleneck representation: $z$ |
| | Embedding | 8 | Categorical feature encoding: $\text{onehot}(x)W_e$ |
| Decoder | Input Layer | 32 | Latent space input: $z$ |
| | Hidden Layer | 128 | Feature decoding: $h_d = \text{ELU}(W_d z + b_d)$ |
| | Output Layer | 5 | Feature reconstruction: $\hat{x}$ |

human-AI cognitive gaps in the latent space.

*Table 4.* Predictor Architectures Configuration

| Component | Layers | Dim/Num of Heads | Description |
|---|---|---|---|
| AI Predictor | Input Layer | 32 | ELU-activated projection into hidden space |
| | Hidden Layer | 128 | ELU transformation of latent features |
| | Output Layer | 2 | Produces class logits for prediction |
| Mask Network | Input Layer | 5 | ELU-activated linear embedding |
| | Attention Layer | 4 | Multi-head self-attention for contextual feature interaction |
| | Output Layer | 32 | Generates masking coefficients |
| Human Predictor | Input Layer | 32 | Takes the masked latent representation as input |
| | Hidden Layer | 128 | ELU transformation of masked latent space |
| | Output Layer | 2 | Produces class logits aligned with experts |

# C. Training Configuration Details

## C.1. Hyperparameter and loss weight selection

All hyperparameters and loss weights were selected via a systematic grid search confined strictly to the training set, ensuring that the independent 20% test set remained untouched throughout model development and thereby preventing data leakage.

Within the 80% training set, we adopted a hold-out validation strategy: 70% of the data were used for model fitting, and the remaining 30% served as a validation subset to evaluate hyperparameter configurations.

- **Learning rate** was searched over the range $[10^{-5}, 10^{-3}]$.

- **Most loss function weights** were searched over a clinically relevant range of $[0.1, 2.0]$.

- **Mask sparsity loss weight**, due to its role as a regularization term requiring finer control to balance sparsity constraints and model performance, was searched over the narrower range of $[10^{-5}, 0.1]$.

Searches were guided by validation AUC, with priority given to configurations demonstrating stable performance (AUC variance $< 0.02$) across three random seeds. The final hyperparameters and loss weights were chosen based on the best validation AUC while ensuring model outputs remained within clinically plausible ranges.

## C.2. Stage-Wise Training Details

The model is trained in four stages: DMAE warm-up, AI predictor training, joint human predictor and mask network training, and fine-tuning. Table 5 specifies the learning rate schedules, batch sizes, and regularization strategies (e.g., gradient clipping) for each phase on the Gitelman syndrome dataset. For instance, the DMAE warm-up phase employs learning rate annealing and early stopping to stabilize latent space initialization. This staged approach balances model complexity and training stability while ensuring task-specific optimization.

*Table 5.* Progressive training strategy

| Phase | Components | Learning Rate | Key Details |
|---|---|---|---|
| DMAE Train | Encoder / Decoder | 1e-4 | • LR annealing
• Early stop
• Gradient clip $\leq 1.0$
• Batch size 16 |
| AI Predictor Train | AI Predictor Network | 1e-4 | • LR annealing
• Early stop
• Gradient clip $\leq 1.0$
• Batch size 16 |
| Human Predictor + Mask Net Train | Human Predictor Network, Mask Network | 1e-4 | • LR annealing
• Early stop
• Gradient clip $\leq 1.0$
• Batch size 16 |
| Fine-Tuning | Full Network | 1e-4 | • Gradient clip $\leq 1.0$
• Batch size 16 |

## C.3. Loss Function Weight in Fine-Tuning Stage

The total training loss combines multiple objectives, including reconstruction, classification, contrastive separation, and cognitive gap minimization. Table 6 defines the weights assigned to each loss component on the Gitelman syndrome dataset, emphasizing the balance between feature reconstruction (dominant in early stages) and rare/common disease separability (enforced via contrastive loss).

*Table 6.* Loss Function Specification

| Loss Type | Weight | Function |
|---|---|---|
| Reconstruction | 1 | Reconstruct input features |
| AI | 1 | Maximize AI prediction accuracy |
| Human | 1 | Align with human diagnoses |
| Mask | 0.001 | Promote sparse attention masks masks |
| Contrastive | 1.5 | Separate rare/common diseases |
| Gap | 1.5 | Reduce human-AI attention gaps |

## C.4. Constraints on Counterfactual Perturbation

To ensure the generated counterfactuals $X'_{cf}$ remain clinically valid, we define the perturbation boundary $\epsilon$ using a multi-layered approach:

**Physiological Reference Intervals.** The reconstructed features are constrained within biological limits derived from standard medical references.

**Expert Validation.** The resulting boundaries were audited by board-certified specialists to confirm they capture transition states between common and rare diagnoses without violating clinical logic.

# D. Details of Prompting LLM and Counterfactual Evaluations

---

**Complete Prompt Template for Clinical Counterfactual Reasoning**

**System Prompt**

- Assume you are a specialist physician (nephrologist/endocrinologist/cardiologist) analyzing a case of [Gitelman syndrome/Acromegaly/Hypertrophic Cardiomyopathy (HCM)].

- **Background Information**: The counterfactual changes in clinical indicators in the following case are generated by perturbing the model along the direction of greatest diagnostic uncertainty as predicted by the physician. This

---

method aims to provide a data-driven alternative perspective that may differ from the initial clinical judgment, helping to correct cognitive anchoring and enabling a more comprehensive assessment of rare diseases.

- The goal is to explain the key diagnostic logic based on the provided changes in indicators and diagnostic probabilities.

- **Important Note for HCM**: The HCM-related indicators (e.g., asymmetric hypertrophy, left ventricular voltage, family history, etc.) are binary variables (0 or 1), where 0 typically indicates negative/normal and 1 indicates positive/abnormal. These are not continuous physiological measurements.

- **The final evaluation should include**:

    - **Summary of Clinical Significance**: Summarize the overall impact of key indicator changes on the diagnosis of [Gitelman syndrome/Acromegaly/HCM].
    - **Explanation of Probability Changes**: Summarize the main reasons for the changes in diagnostic probabilities.
    - **Clinical Value of 'nan' Perturbations**: Summarize the significance of perturbing 'nan' (missing) values for prompting further tests and diagnosing rare diseases.
    - **Value of Counterfactual Simulation**: Briefly describe, based on its generation mechanism (perturbing in the direction of greatest uncertainty to correct cognitive anchoring), how this simulation helps clinical diagnosis, especially in avoiding premature exclusion of rare diseases.

### User Prompt

- The key indicator changes for case $\{i+1\}$ are as follows, where the original value of 'nan' indicates that the test was not performed:

- **[Disease-specific indicators]**: e.g., for Gitelman: Urine Potassium, Bicarbonate, Serum Potassium, High Blood Pressure, pH; for Acromegaly: IGF-1, Serum GH, OGTT-GH; for HCM: Asymmetric Hypertrophy, Low/High Left Ventricular Voltage, Family History, Amyloid Deposition, Sarcomere Gene Mutation, LVOTO, SAM, TTR Gene Mutation.

- **Diagnostic probability changes**:

    - Original (Common Disease/Rare Disease): $\{p_{common}\}/\{p_{rare}\}$
    - After Perturbation (Common Disease/Rare Disease): $\{p_{common\_perturb}\}/\{p_{rare\_perturb}\}$

- Please provide a detailed process analysis and result evaluation based on the above data and your medical knowledge.

### LLM Response Analysis and Clinical Evaluation Results

**Case 1: Acromegaly**

- Initially relying solely on IGF-1 and basal GH levels may not clarify the diagnosis (both may be at critical values or nonspecific), leading to similar probabilities for the rare disease (acromegaly) and common diseases.

- After perturbation, the OGTT-GH suppression test result directly confirms the diagnosis. The failure of GH to suppress during OGTT—a key pathological feature of acromegaly—exhibits extremely high specificity, effectively ruling out other common diseases.

- Omission of the critical OGTT-GH suppression test introduces diagnostic uncertainty and may lead to missed acromegaly.

- **Conclusion**: Abnormal results from the OGTT-GH suppression test are decisive evidence for acromegaly diagnosis. Their absence leads to diagnostic ambiguity, while supplementation significantly improves diagnostic specificity.

**Case 2: Gitelman Syndrome (Ambiguity Correction)**

- In the original data, clinicians may exclude Gitelman syndrome based solely on normal serum potassium. After counterfactual perturbation, a bicarbonate level of $38\ mmol/L$ and pH 7.6 clearly indicate metabolic alkalosis, prompting reassessment of renal tubular dysfunction.

- **Value of Counterfactual Simulation**: Overcoming uncertainty and correcting cognitive anchoring regarding "normal serum potassium."

- **Conclusion**: This case demonstrates that counterfactual simulation, by supplementing critical evidence of metabolic alkalosis, helps clinicians overcome cognitive limitations of "hypokalemia" and "common disease priority."

**Case 3: Gitelman Syndrome (Renal Wasting)**

- **Core Feature**: Renal potassium wasting. Elevated urine potassium ($> 20\ mmol/24h$), combined with hypokalemia (serum potassium $2.2\ mmol/L$), strongly supports renal tubular dysfunction.

- **Mechanism**: Metabolic alkalosis is driven by renal hydrogen ion loss (via $Na^+ - H^+$ exchange compensating for hypovolemia) and RAAS activation.

- **Conclusion**: This case, by perturbing "nan" values of urine potassium and $HCO_3^-$, reveals the critical paradox of Gitelman syndrome (hypokalemia + hyperuricosuria + metabolic alkalosis + no hypertension).

**Descrption:**
In the presented cases:

1. The true labels represent the actual disease status recorded in clinical practice.

2. For each indicator, the value before the arrow is the patient's actual test result (where "nan" indicates that the patient did not undergo that particular test), and the value after the arrow is the generated counterfactual indicator result.

3. For missing indicators, the $\Delta$ change is calculated as the difference between the mean value of that indicator in the dataset and the counterfactual data.

4. The changes in prediction probabilities are obtained from a trained accurate AI model.

5. **Example**: When original tests were conducted, the model predicted the probability of a common disease to be 0.7770 and that of a rare disease to be 0.2230. After counterfactual perturbation, the rare disease probability increased to 0.8703, intuitively showing the reversing effect of supplementing key indicators.

# E. Extended Latent Space Visualization

In this section, we present the visualizations of the latent space manifold derived from the Gitelman syndrome cohort (Figure 5). These plots provide a visual confirmation of the reasoning principles discussed in the main text. Panel (a) represents the structural class distribution in the latent space, illustrating the model's ability to cluster clinically relevant phenotypes. Panel (b) maps the mask distribution to reflect clinician attention, where mask=1 signifies regions of high clinical relevance. Panel (c) quantifies clinician uncertainty intensity (increasing from dark to light), highlighting critical areas where human prediction is most volatile and susceptible to cognitive bias.

# F. Latent Space Visualization with Ablation Study

We conduct an ablation study to evaluate the necessity of each loss term in our model's total loss function. Specifically, we visualize the distribution of the latent space when individually removing each loss component during fine-tuning (prior to fine-tuning, each component of our model, including DMAE, AI predictor, mask net and human predictor, is first trained in

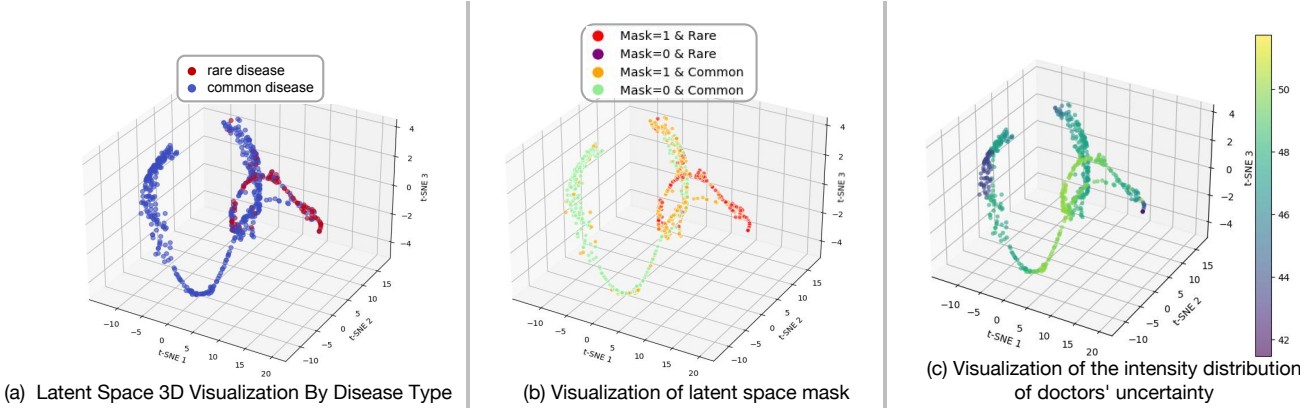

(a) Latent Space 3D Visualization By Disease Type

(b) Visualization of latent space mask

(c) Visualization of the intensity distribution of doctors' uncertainty

*Figure 5.* Latent space visualization by disease type, clinician attention, and diagnostic uncertainty.

stages with its corresponding loss function). As shown in Figure 6, Our findings indicate that the removal of the contrastive loss, gap loss, or reconstruction loss degrades the quality of the latent space representation, thereby impairing the model's ability to discriminate between similar samples.

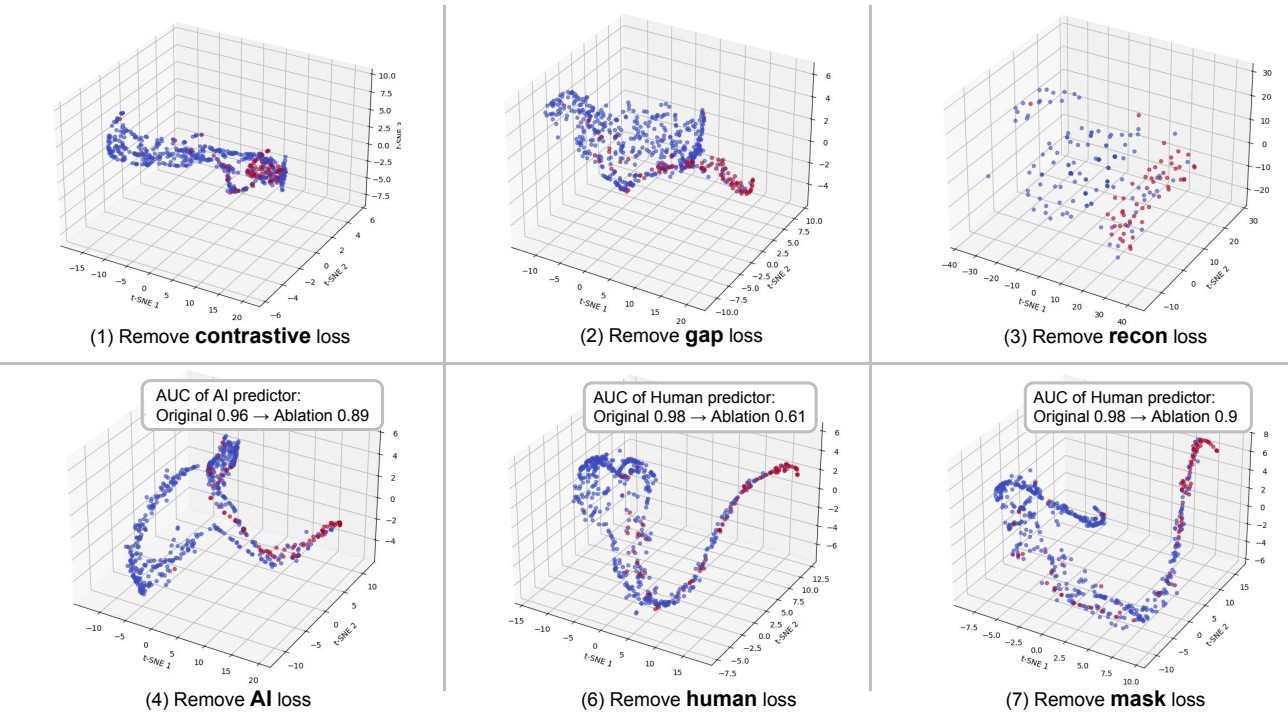

(1) Remove **contrastive** loss

(2) Remove **gap** loss

(3) Remove **recon** loss

(4) Remove **AI** loss

(6) Remove **human** loss

(7) Remove **mask** loss

*Figure 6.* Ablation study: loss function removal impact on latent space and model performance.

In contrast, removal of the AI prediction loss, AI prediction loss or mask regularization loss impairs the performance of the AI predictor or human predictor, as depicted by the AUC changes in the figure, underscoring the indispensable role of each loss component in maintaining model effectiveness.

## G. Detailed Predictive Performance and Ablation Results

This section presents the complete experimental evaluation and ablation analysis. Table 7 details the AUC results (mean ± standard deviation over 10 runs) for both AI and human predictors across all seven clinical cohorts.

*Table 7.* AUC metrics and ablations across seven datasets (10-run avg $\pm$ std).

| | GPA | | IHPRF3 | | CCMS | | Multiple Myeloma | |
| | AI | Human | AI | Human | AI | Human | AI | Human |
|---|---|---|---|---|---|---|---|---|
| Original | **0.98±0.02** | **0.97±0.01** | **0.97±0.03** | **0.98±0.01** | **1.00±0.00** | **1.00±0.00** | **0.97±0.02** | **0.97±0.01** |
| No AI loss | 0.90±0.03 | N/A | 0.93±0.04 | N/A | 0.98±0.02 | N/A | 0.91±0.06 | N/A |
| No human loss | N/A | 0.82±0.04 | N/A | 0.89±0.09 | N/A | 0.93±0.06 | N/A | 0.85±0.07 |
| No mask loss | N/A | 0.93±0.03 | N/A | 0.95±0.03 | N/A | 0.98±0.02 | N/A | 0.93±0.03 |

| | Gitelman | | Acromegaly | | HCM | |
| | AI | Human | AI | Human | AI | Human |
|---|---|---|---|---|---|---|
| Original | **0.96±0.01** | **0.98±0.01** | **0.99±0.01** | **0.98±0.03** | **0.96±0.01** | **0.97±0.01** |
| No AI loss | 0.89±0.08 | N/A | 0.96±0.02 | N/A | 0.86±0.05 | N/A |
| No human loss | N/A | 0.61±0.14 | N/A | 0.87±0.07 | N/A | 0.73±0.12 |
| No mask loss | N/A | 0.90±0.08 | N/A | 0.94±0.05 | N/A | 0.92±0.02 |

## H. Full List of Expert-Reviewed Counterfactual Examples

This appendix reports the complete set of expert-reviewed counterfactual examples. Each case lists the counterfactual perturbation, the model prediction before and after perturbation, and the physician evaluation.

> Counterfactual data explanation:
> In clinical practice, rare-disease diagnosis is often distorted by missing key examination indicators, causing physicians to misclassify patients as having common diseases. This study uses counterfactual case analysis to show how supplementing key indicators changes diagnostic outcomes, with the goal of improving physicians' recognition of rare-disease features and supporting precise decision-making.
> The first sheet (sheet 1–label flip) shows how missing examination indicators would change if the diagnosis of a patient with a common disease were shifted to a rare disease. The table covers three rare-disease groups: Gitelman syndrome, acromegaly, and HCM, with three cases provided for each group.
> The second sheet (sheet 2–uncertainty) focuses on the most uncertain parts of physician decision-making, changes examination indicator results, and shows how disease probabilities should change accordingly. This sheet provides three cases from the Gitelman dataset.
> The third sheet (sheet 3–low confident) addresses situations in which the patient's original input features contain missing values, substantially overlap with common-disease features, and yield a low-confidence AI prediction favoring common disease (for example, a classification probability close to 0.5). In such cases, our model generates counterfactual samples to infer the missing features.
> The fourth sheet (sheet 4–discrepancy) addresses situations in which the AI prediction differs from the clinician's diagnosis. Our model generates flip samples to reveal differences in decision logic.
> Supplementary note: the newly added cases also cover target diseases from four public datasets, including HMS-GPA (GPA), LIRICAL-IHPRF3, MME-CCMS, and MIMIC-IV-Rare (multiple myeloma), to further illustrate counterfactual results under label-flip, diagnostic-uncertainty, low-confidence, and AI-physician discrepancy settings.

> In the cases shown below:
> 1. The true label is the actual disease status recorded in the clinical record.
> 2. For each indicator, the value before the arrow is the patient's actual examination result, and the value after the arrow is the generated counterfactual result. For missing indicators, the change in $\Delta$ is calculated as the difference between the dataset mean of that indicator and the counterfactual data; for other indicators, the change in $\Delta$ is calculated as the difference between the original data and the counterfactual data.
> 3. The predicted probability changes come from a trained precision AI model. A higher score indicates a higher likelihood. Each case shows both the AI prediction for the patient's original examinations and the AI prediction under the counterfactual result. Using the first case below as an example:
> Under the original examinations, the model predicted a 0.7770 probability of common disease and 0.2230 for rare disease. After the counterfactual perturbation, the probability of common disease dropped sharply to 0.1297, while the probability of rare disease rose to 0.8703, directly illustrating how supplementing key indicators can reverse diagnostic tendency. This case shows that reasonable completion of key missing indicators can substantially change AI diagnostic tendency through counterfactual reasoning, providing quantitative reference support for clinicians to identify potential rare diseases.

Features in the Gitelman dataset include urine potassium (urine_potassium), bicarbonate, serum potassium (serum_potassium), high blood pressure (high_blood_pressure), and PH.

Features in the acromegaly dataset include the minimum growth hormone value during the oral glucose tolerance test (OGTT-GH_min), serum growth hormone level (serum_GH), and insulin-like growth factor 1 level (IGF-1).

Features in the HCM dataset include SAM (systolic anterior motion of the mitral valve), TTR Gene Mutation, Amyloid Deposition, Sarcomere Gene Mutation, Family History, High Left Ventricular Voltage, Asymmetric Hypertrophy, and Low Left Ventricular Voltage.

Features in the HMS-GPA dataset include Age, Sex_bool, hemoptysis, PR3-ANCA, c-ANCA, and otitis media.

Features in the LIRICAL-IHPRF3 dataset include Severe muscular hypotonia, Developmental regression, Severe global developmental delay, and Intellectual disability.

Features in the MME-CCMS dataset include Posterior rib gap, Pierre-Robin sequence, Cleft palate, Global developmental delay, Scoliosis, and High palate.

Features in the MIMIC-IV-Rare multiple myeloma dataset include anemia, fatigue, back pain, bone pain, lytic bone lesions, hypercalcemia, renal insufficiency, and monoclonal gammopathy.

## H.1. Label-Flip Counterfactuals

1. **Gitelman syndrome, Case 1.** True label: common disease. Perturbation: urine_potassium: nan $\rightarrow$ 211.7418 ($\Delta \uparrow$ 164.2331); bicarbonate: nan $\rightarrow$ 38.0000 ($\Delta \uparrow$ 11.1840); serum_potassium: 2.4000 $\rightarrow$ 2.4000; PH: 7.4600 $\rightarrow$ 7.4600; high_blood_pressure: 1.0000 $\rightarrow$ 1.0000. Prediction change: common disease 0.7770, rare disease 0.2230 $\rightarrow$ common disease 0.1297, rare disease 0.8703. *Physician evaluation:* By completing urine potassium and bicarbonate, this counterfactual redirects the clinician's attention from surface hypokalemia to the core pathophysiology of renal potassium wasting with metabolic alkalosis. Its primary value lies in quantifying the effect of missing electrolyte and blood-gas evidence and reducing delayed recognition of Gitelman syndrome caused by incomplete routine workup.

2. **Gitelman syndrome, Case 2.** True label: common disease. Perturbation: urine_potassium: nan $\rightarrow$ 157.5374 ($\Delta \uparrow$ 110.4591); serum_potassium: nan $\rightarrow$ 0.3046 ($\Delta \downarrow$ 2.6300); PH: 7.4400 $\rightarrow$ 7.4400; bicarbonate: 28.9000 $\rightarrow$ 28.9000; high_blood_pressure: 1.0000 $\rightarrow$ 1.0000. Prediction change: common disease 0.7045, rare disease 0.2955 $\rightarrow$ common disease 0.0001, rare disease 0.9999. *Physician evaluation:* This case shows that the key issue in hypokalemia is not simply whether potassium is low, but whether renal versus non-renal loss has been adequately characterized. By structuring the missing urine potassium and related biochemical results, the simulation helps break a common-disease-first bias and supports a more complete differential diagnosis.

3. **Gitelman syndrome, Case 3.** True label: common disease. Perturbation: urine_potassium: nan $\rightarrow$ 210.5626 ($\Delta \uparrow$ 163.0540); PH: nan $\rightarrow$ 7.2001 ($\Delta \downarrow$ 0.2356); serum_potassium: 2.7000 $\rightarrow$ 2.7000; bicarbonate: 28.4000 $\rightarrow$ 28.4000; high_blood_pressure: 0.0000 $\rightarrow$ 0.0000. Prediction change: common disease 0.7256, rare disease 0.2744 $\rightarrow$ common disease 0.1223, rare disease 0.8777. *Physician evaluation:* By restoring urine potassium and PH, this scenario moves diagnostic reasoning away from nonspecific hypokalemia and toward a renal tubular mechanism more consistent with Gitelman syndrome. Its value lies in reminding clinicians that even atypical presentations may conceal rare disease when the most informative tests have not been completed.

4. **Acromegaly, Case 1.** True label: common disease. Perturbation: OGTT-GH_min: nan $\rightarrow$ 118.4337 ($\Delta \uparrow$ 118.4337); serum_GH: 0.3000 $\rightarrow$ 0.3000; IGF-1: 174.0000 $\rightarrow$ 174.0000. Prediction change: common disease 0.7956, rare disease 0.2044 $\rightarrow$ common disease 0.0003, rare disease 0.9997. *Physician evaluation:* This simulation effectively reconstructs the OGTT-GH suppression test as the diagnostic gold standard, converting a judgment based on static hormone levels into a more definitive interpretation of acromegaly. Its main value is in quantifying how much diagnostic uncertainty can be created by omission of dynamic functional testing.

5. **Acromegaly, Case 2.** True label: common disease. Perturbation: OGTT-GH_min: nan $\rightarrow$ 11.9131 ($\Delta \uparrow$ 11.9131); serum_GH: 2.4000 $\rightarrow$ 2.4000; IGF-1: 238.0000 $\rightarrow$ 238.0000. Prediction change: common disease 0.6101, rare disease 0.3899 $\rightarrow$ common disease 0.4585, rare disease 0.5415. *Physician evaluation:* By restoring the OGTT result, this case shifts the clinician's reasoning from nonspecific endocrine abnormality toward the underlying pathophysiology of failed GH suppression. It helps correct overreliance on isolated GH or IGF-1 values and keeps the diagnostic logic more transparent and objective.

6. **Acromegaly, Case 3.** True label: common disease. Perturbation: OGTT-GH_min: nan $\rightarrow$ 34.4365 ($\Delta \uparrow$ 34.4365); serum_GH: 1.8000 $\rightarrow$ 1.8000; IGF-1: 210.0000 $\rightarrow$ 210.0000. Prediction change: common disease 0.6947, rare disease 0.3053 $\rightarrow$ common disease 0.2310, rare disease 0.7690. *Physician evaluation:* it simulates the "gold standard" OGTT-GH suppression test, transforming an ambiguous suspicion into a definitive diagnosis. Its primary value lies in quantifying

the weight of missing evidence, effectively breaking clinical "wait-and-see" stalemates and preventing misdiagnosis rooted in incomplete data.

7. **Hypertrophic cardiomyopathy, Case 1.** True label: ATTR. Perturbation: SAM: $0.0000 \rightarrow 0.9661$; low left ventricular voltage: $0.0000 \rightarrow 0.9569$; sarcomere gene mutation: $0.0000 \rightarrow 0.8288$; family history: $0.0000 \rightarrow 0.7118$; asymmetric hypertrophy: $0.0000 \rightarrow 0.6103$; high left ventricular voltage: $0.0000 \rightarrow 0.4346$; TTR gene mutation: $1.0000 \rightarrow 0.8867$; amyloid deposition: $1.0000 \rightarrow 0.8878$. Prediction change: ATTR 0.6639, HCM 0.3361 $\rightarrow$ ATTR 0.0983, HCM 0.9017. *Physician evaluation:* This counterfactual redirects clinical attention from isolated imaging or genetic findings to the broader pathophysiologic pattern supporting HCM. By structuring the relative weight of the major features, it improves transparency when distinguishing HCM from ATTR and reduces overdependence on any single clue.

8. **Hypertrophic cardiomyopathy, Case 2.** True label: ATTR. Perturbation: SAM: $0.0000 \rightarrow 0.9751$; low left ventricular voltage: $0.0000 \rightarrow 0.9682$; sarcomere gene mutation: $0.0000 \rightarrow 0.8580$; family history: $0.0000 \rightarrow 0.7401$; high left ventricular voltage: $0.0000 \rightarrow 0.5006$; asymmetric hypertrophy: $1.0000 \rightarrow 0.6934$; amyloid deposition: $1.0000 \rightarrow 0.9057$; TTR gene mutation: $1.0000 \rightarrow 0.9161$. Prediction change: ATTR 0.6562, HCM 0.3438 $\rightarrow$ ATTR 0.1003, HCM 0.8997. *Physician evaluation:* This case helps clinicians understand the relative contribution of genetics, family history, and echocardiographic findings in HCM. Its main value lies in turning a complex multimodal differential into a more interpretable decision pathway for challenging cardiomyopathy cases.

9. **Hypertrophic cardiomyopathy, Case 3.** True label: ATTR. Perturbation: SAM: $0.0000 \rightarrow 0.9885$; TTR gene mutation: $0.0000 \rightarrow 0.9639$; amyloid deposition: $0.0000 \rightarrow 0.9445$; sarcomere gene mutation: $0.0000 \rightarrow 0.9104$; family history: $0.0000 \rightarrow 0.8474$; high left ventricular voltage: $0.0000 \rightarrow 0.6527$; asymmetric hypertrophy: $1.0000 \rightarrow 0.8428$; low left ventricular voltage: $1.0000 \rightarrow 0.9867$. Prediction change: ATTR 0.6754, HCM 0.3246 $\rightarrow$ ATTR 0.1110, HCM 0.8890. *Physician evaluation:* This simulation shows how HCM-specific findings can regain diagnostic priority even in the presence of overlapping ATTR-like evidence. For clinicians, its practical value is in preventing overinterpretation of partially shared features and promoting more targeted confirmatory evaluation.

10. **GPA, Case 1.** True label: common disease. Perturbation: hemoptysis: NA $\rightarrow$ NA; PR3-ANCA: NA $\rightarrow$ 1; c-ANCA: NA $\rightarrow$ 1; otitis media: $1 \rightarrow 1$; age: $52 \rightarrow 52$; sex_bool: $1 \rightarrow 1$. Prediction change: common disease 0.6840, rare disease 0.3160 $\rightarrow$ common disease 0.2590, rare disease 0.7410. *Physician evaluation:* It completes two high-yield serologic findings, PR3-ANCA and c-ANCA, and thereby turns an otherwise ambiguous suspicion of common respiratory or local ENT disease into a more defensible GPA-oriented interpretation. Its primary value lies in quantifying how much missing immunologic evidence can distort diagnostic direction and helping clinicians move beyond a passive "treat as common disease and wait" strategy.

11. **GPA, Case 2.** True label: common disease. Perturbation: hemoptysis: NA $\rightarrow$ 1; PR3-ANCA: NA $\rightarrow$ NA; c-ANCA: NA $\rightarrow$ NA; otitis media: NA $\rightarrow$ 1; age: $43 \rightarrow 43$; sex_bool: $0 \rightarrow 0$. Prediction change: common disease 0.7312, rare disease 0.2688 $\rightarrow$ common disease 0.3860, rare disease 0.6140. *Physician evaluation:* This scenario shows that a cross-organ symptom combination can itself justify escalation of diagnostic thinking. By reconstructing hemoptysis and otitis media, it shifts the clinician's focus from local infection toward a systemic vasculitic process even before the full serologic workup is available.

12. **IHPRF3, Case 1.** True label: common disease. Perturbation: severe muscular hypotonia: NA $\rightarrow$ 1; developmental regression: NA $\rightarrow$ NA; severe global developmental delay: $1 \rightarrow 1$; intellectual disability: $1 \rightarrow 1$. Prediction change: common disease 0.7425, rare disease 0.2575 $\rightarrow$ common disease 0.3120, rare disease 0.6880. *Physician evaluation:* By restoring severe muscular hypotonia, the system reorients the case from a vague developmental-delay picture toward a more syndromic neurodevelopmental interpretation. Its main value lies in quantifying the leverage of one missing core phenotype and reducing the risk that incomplete charting suppresses suspicion of a rare syndrome.

13. **IHPRF3, Case 2.** True label: common disease. Perturbation: severe global developmental delay: NA $\rightarrow$ NA; severe muscular hypotonia: NA $\rightarrow$ NA; developmental regression: NA $\rightarrow$ 1; intellectual disability: $1 \rightarrow 1$. Prediction change: common disease 0.7010, rare disease 0.2990 $\rightarrow$ common disease 0.3970, rare disease 0.6030. *Physician evaluation:* This counterfactual elevates developmental regression from an unrecorded clue to a central driver of interpretation, shifting the clinician's attention from static phenotype description to disease-course reasoning. For junior clinicians or non-specialists, that structured reframing has clear teaching value; for experienced neurodevelopmental specialists, however, the case is somewhat less novel and functions more as quantitative reinforcement than as a major new insight.

14. **CCMS, Case 1.** True label: common disease. Perturbation: posterior rib gap: NA → NA; Pierre-Robin sequence: NA → 1; cleft palate: 1 → 1; high palate: NA → NA; scoliosis: 0 → 0. Prediction change: common disease 0.7674, rare disease 0.2326 → common disease 0.3080, rare disease 0.6920. *Physician evaluation:* By introducing Pierre-Robin sequence, the simulation converts what might otherwise be viewed as an isolated oral anomaly into a more coherent syndromic craniofacial pattern. Its value lies in helping clinicians identify which single structural feature is most worth confirming first so that examination and imaging become more diagnostically directed.

15. **CCMS, Case 2.** True label: common disease. Perturbation: cleft palate: NA → 1; posterior rib gap: NA → NA; global developmental delay: 1 → 1; high palate: NA → NA; Pierre-Robin sequence: NA → 1. Prediction change: common disease 0.6882, rare disease 0.3118 → common disease 0.2790, rare disease 0.7210. *Physician evaluation:* This case shows that only a few anatomically coherent structural abnormalities may be sufficient to reshape the entire syndromic interpretation. By reconstructing cleft palate and Pierre-Robin sequence, it shifts attention away from superficial developmental delay and toward the structural framework of CCMS.

16. **Multiple myeloma, Case 1.** True label: common disease. Perturbation: anemia: 1 → 1; back pain: 1 → 1; lytic bone lesions: NA → 1; monoclonal gammopathy: NA → 1; hypercalcemia: NA → NA; renal insufficiency: 1 → 1. Prediction change: common disease 0.7418, rare disease 0.2582 → common disease 0.2190, rare disease 0.7810. *Physician evaluation:* This simulation restores lytic bone lesions and monoclonal gammopathy, transforming a case that could easily remain hidden within common chronic-disease patterns into a clearer plasma-cell malignancy profile. Its main value lies in quantifying how much delayed recognition can be driven by omission of the most informative tests.

17. **Multiple myeloma, Case 2.** True label: common disease. Perturbation: fatigue: 1 → 1; bone pain: 1 → 1; compression fracture: NA → NA; monoclonal gammopathy: NA → 1; anemia: NA → NA; chronic kidney disease: 1 → 1. Prediction change: common disease 0.7026, rare disease 0.2974 → common disease 0.3820, rare disease 0.6180. *Physician evaluation:* This case demonstrates that a single high-specificity clue can meaningfully redirect diagnosis without waiting for the full CRAB picture to emerge. For general internists or non-hematologists, that reminder is useful and corrective; for experienced hematologists, however, the case is somewhat less illuminating and may function more as structured confirmation of existing judgment than as a major new insight.

## H.2. Diagnostic-Uncertainty Counterfactuals

1. **Gitelman syndrome, Case 1.** True label: common disease. Perturbation: urine_potassium: nan → 123.1027 ($\Delta \uparrow$ 75.8599); bicarbonate: nan → 36.9235 ($\Delta \uparrow$ 10.0874); serum_potassium: 2.2000 → 2.2000; PH: 7.4350 → 7.4350; high_blood_pressure: 0.0000 → 0.0000. Prediction change: common disease 0.7262, rare disease 0.2738 → common disease 0.3834, rare disease 0.6166. *Physician evaluation:* By completing urine potassium and bicarbonate, this case pulls the reasoning process away from generic hypokalemia and back toward renal tubular pathophysiology. Its value lies in showing that the most uncertain parts of diagnosis are often exactly the tests that deserve highest priority.

2. **Gitelman syndrome, Case 2.** True label: common disease. Perturbation: urine_potassium: nan → 10.2127 ($\Delta \downarrow$ 37.0301); serum_potassium: nan → 3.9243 ($\Delta \uparrow$ 0.9877); PH: 7.3670 → 7.3670; bicarbonate: 24.3000 → 24.3000; high_blood_pressure: 1.0000 → 1.0000. Prediction change: common disease 0.7496, rare disease 0.2504 → common disease 0.7589, rare disease 0.2411. *Physician evaluation:* This case reminds clinicians that counterfactual completion does not always increase rare-disease probability; sometimes the main value is greater confidence in excluding a rare disease. That too is clinically useful because it reduces overinterpretation under incomplete evidence.

3. **Gitelman syndrome, Case 3.** True label: common disease. Perturbation: urine_potassium: nan → 122.7564 ($\Delta \uparrow$ 75.5136); bicarbonate: nan → 36.9033 ($\Delta \uparrow$ 10.0672); serum_potassium: 2.7000 → 2.7000; PH: 7.4500 → 7.4500; high_blood_pressure: 0.0000 → 0.0000. Prediction change: common disease 0.7199, rare disease 0.2801 → common disease 0.3799, rare disease 0.6201. *Physician evaluation:* By reconstructing urine potassium and bicarbonate, the simulation redirects attention from surface hypokalemia to the mechanism most consistent with Gitelman syndrome. Its primary value is in correcting the assumption that a conclusion can be made before the most informative biochemical tests are completed.

4. **GPA, Case 1.** True label: common disease. Perturbation: hemoptysis: NA → NA; c-ANCA: NA → 1; PR3-ANCA: 0 → 0; otitis media: 1 → 1; age: 58 → 58; sex_bool: 1 → 1. Prediction change: common disease 0.5610, rare disease 0.4390 → common disease 0.3930, rare disease 0.6070. *Physician evaluation:* This counterfactual isolates c-ANCA as the single most useful next test, which makes the diagnostic process resemble real staged clinical workup rather than an

unrealistically complete data reveal. Its value lies in showing clinicians exactly which missing item is most likely to reduce uncertainty and break anchoring in common ENT or infectious explanations.

5. **Multiple myeloma, Case 2.** True label: common disease. Perturbation: anemia: $1 \rightarrow 1$; fatigue: $1 \rightarrow 1$; monoclonal gammopathy: NA $\rightarrow$ 1; lytic bone lesions: NA $\rightarrow$ NA; back pain: NA $\rightarrow$ NA. Prediction change: common disease 0.6032, rare disease 0.3968 $\rightarrow$ common disease 0.4520, rare disease 0.5480. *Physician evaluation:* By restoring only monoclonal gammopathy, the system lets clinicians see how much one decisive laboratory result can alter diagnostic probability. This is particularly valuable for breaking the habit of dismissing myeloma too early when the presenting symptoms appear overly common, and it keeps the diagnostic logic transparent by tying the shift to one clearly interpretable piece of evidence.

### H.3. Low-Confidence Counterfactuals

1. **Acromegaly syndrome.** True label: rare disease. Perturbation: OGTT-GH_min: nan $\rightarrow$ 298.2090 ($\Delta \uparrow$ 298.2090); serum_GH: 4.3000 $\rightarrow$ 4.3000; IGF-1: 272.0000 $\rightarrow$ 272.0000. Prediction change: common disease 0.4738, rare disease 0.5262 $\rightarrow$ common disease 0.0000, rare disease 1.0000. *Physician evaluation:* This counterfactual effectively restores the single most decisive missing piece in acromegaly confirmation, turning a near-threshold judgment into a much clearer diagnosis. Its core value is to show that low confidence does not imply low risk; often it simply means that the gold-standard test has not yet been obtained.

2. **IHPRF3 syndrome.** True label: rare disease. Perturbation: severe muscular hypotonia: NA $\rightarrow$ 1; developmental regression: NA $\rightarrow$ NA; severe global developmental delay: $1 \rightarrow 1$; intellectual disability: $1 \rightarrow 1$. Prediction change: common disease 0.5230, rare disease 0.4770 $\rightarrow$ common disease 0.2590, rare disease 0.7410. *Physician evaluation:* This scenario captures a classic near-threshold case in which the problem is not that the disease lacks syndromic structure, but that one core phenotype has not yet been documented. By restoring severe muscular hypotonia, the system turns a low-confidence prediction into a much clearer rare-disease pattern and quantifies the impact of that missing feature.

3. **CCMS.** True label: rare disease. Perturbation: Pierre-Robin sequence: NA $\rightarrow$ 1; posterior rib gap: NA $\rightarrow$ NA; cleft palate: $1 \rightarrow 1$; high palate: $1 \rightarrow 1$; global developmental delay: $0 \rightarrow 0$. Prediction change: common disease 0.5140, rare disease 0.4860 $\rightarrow$ common disease 0.2920, rare disease 0.7080. *Physician evaluation:* By adding Pierre-Robin sequence, this counterfactual transforms a low-confidence overlap case into a more coherent syndromic craniofacial presentation. For clinicians, its value lies in shifting the next step from passive observation to targeted confirmation of structural features; however, for specialists who already manage craniofacial syndromes regularly, the incremental insight may be somewhat modest and function mainly as quantitative support for familiar reasoning.

### H.4. AI-Physician Discrepancy Counterfactuals

1. **Gitelman syndrome.** True label: common disease. Perturbation: bicarbonate: NA $\rightarrow$ 38 ($\Delta \uparrow$ 11.1677); PH: 7.43 $\rightarrow$ 7.6 ($\Delta \uparrow$ 0.1700); serum_potassium: 4.1 $\rightarrow$ 4.1; urine_potassium: 82 $\rightarrow$ 82; high_blood_pressure: 0.0000 $\rightarrow$ 0.0000. Prediction of whether the original data indicate rare disease: AI yes, physician no. Prediction of whether the counterfactual result indicates rare disease: AI yes, physician yes. *Physician evaluation:* By reconstructing the "missing piece" of the biochemical puzzle (Bicarbonate), it shifts the clinician's focus from surface symptoms to underlying pathophysiology. This effectively corrects anchoring bias and ensures diagnostic logic remains transparent and objective.

2. **GPA.** True label: common disease. Perturbation: hemoptysis: NA $\rightarrow$ NA; PR3-ANCA: NA $\rightarrow$ 1; age: 58 $\rightarrow$ 58; sex_bool: $1 \rightarrow 1$; c-ANCA: $0 \rightarrow 0$; otitis media: $0 \rightarrow 0$. Prediction of whether the original data indicate rare disease: AI yes, physician no. Prediction of whether the counterfactual result indicates rare disease: AI yes, physician yes. *Physician evaluation:* By simulating critical scenarios such as PR3-ANCA positivity in diagnostic "gray zones," the system offers structured alternative hypotheses. This pushes physicians to evaluate more severe clinical possibilities, ensuring that high-risk rare diseases are not mistakenly dismissed as common infections.

3. **CCMS.** True label: common disease. Perturbation: cleft palate: $1 \rightarrow 1$; high palate: $1 \rightarrow 1$; posterior rib gap: NA $\rightarrow$ NA; Pierre-Robin sequence: NA $\rightarrow$ 1; scoliosis: NA $\rightarrow$ NA. Prediction of whether the original data indicate rare disease: AI yes, physician no. Prediction of whether the counterfactual result indicates rare disease: AI yes, physician yes. *Physician evaluation:* This case reconstructs Pierre-Robin sequence as the missing structural clue that moves the interpretation from isolated craniofacial anomaly toward syndromic disease. Its importance lies in showing that disagreement between

AI and clinicians often comes not from the final label itself, but from different weight assigned to one critical missing phenotype.

4. **Multiple myeloma.** True label: common disease. Perturbation: anemia: $1 \rightarrow 1$; bone pain: $1 \rightarrow 1$; monoclonal gammopathy: NA $\rightarrow$ 1; lytic bone lesions: NA $\rightarrow$ NA; hypercalcemia: NA $\rightarrow$ NA. Prediction of whether the original data indicate rare disease: AI yes, physician no. Prediction of whether the counterfactual result indicates rare disease: AI yes, physician yes. *Physician evaluation:* By introducing monoclonal gammopathy, the system redirects the clinician's focus away from common explanations for anemia and bone pain and back toward the pathophysiology of plasma-cell malignancy. The main value here is not broad hypothesis generation, but precise identification of which single additional test is most likely to change real clinical decision-making when AI and clinician judgment diverge.

## I. Illustrative Label-Flip Case Comparisons with Baselines

To contextualize the label-flip counterfactuals, we compare our generated cases with CFVAE and REVISE using representative examples. CFVAE and REVISE are from our paper-aligned reimplementations. Because each baseline has its own classifier, absolute probabilities are not directly comparable across methods; the relevant signal is the within-method before/after change and the clinical pattern of feature edits.

**Gitelman Syndrome.** **Original input:** `urine_potassium`: NA; PH: NA; `serum_potassium`: 2.7; bicarbonate: 28.4; `high_blood_pressure`: 0.

**Ours.** Counterfactual output: `urine_potassium`: NA $\rightarrow$ 210.5626; PH: NA $\rightarrow$ 7.2001; `serum_potassium`: 2.7 $\rightarrow$ 2.7; bicarbonate: 28.4 $\rightarrow$ 28.4; `high_blood_pressure`: 0 $\rightarrow$ 0. Probability change: rare disease 0.2744 $\rightarrow$ 0.8777.

**CFVAE.** Counterfactual output: `urine_potassium`: NA $\rightarrow$ 136.00; PH: NA $\rightarrow$ 7.4561; `serum_potassium`: 2.7 $\rightarrow$ 3.15; bicarbonate: 28.4 $\rightarrow$ 31.45; `high_blood_pressure`: 0 $\rightarrow$ 0.49. Probability change: rare disease 0.1983 $\rightarrow$ 0.8944.

**REVISE.** Counterfactual output: `urine_potassium`: NA $\rightarrow$ 74.68; PH: NA $\rightarrow$ 7.4459; `serum_potassium`: 2.7 $\rightarrow$ 2.92; bicarbonate: 28.4 $\rightarrow$ 28.54; `high_blood_pressure`: 0 $\rightarrow$ 0.03. Probability change: rare disease 0.0785 $\rightarrow$ 0.4574.

*Takeaway.* Our counterfactual is more focused on the clinically central renal-tubular evidence and achieves a decisive label flip with less collateral rewriting. CFVAE is more aggressive but broader, whereas REVISE is more conservative and does not flip the case as strongly.

**Acromegaly.** **Original input:** `OGTT-GH_min`: NA; `serum_GH`: 0.3; IGF-1: 174.

**Ours.** Counterfactual output: `OGTT-GH_min`: NA $\rightarrow$ 118.4337; `serum_GH`: 0.3 $\rightarrow$ 0.3; IGF-1: 174 $\rightarrow$ 174. Probability change: rare disease 0.2044 $\rightarrow$ 0.9997.

**CFVAE.** Counterfactual output: `OGTT-GH_min`: NA $\rightarrow$ 67.04; `serum_GH`: 0.3 $\rightarrow$ 84.10; IGF-1: 174 $\rightarrow$ 570.84. Probability change: rare disease 0.4853 $\rightarrow$ 0.5165.

**REVISE.** Counterfactual output: `OGTT-GH_min`: NA $\rightarrow$ 71.01; `serum_GH`: 0.3 $\rightarrow$ 90.03; IGF-1: 174 $\rightarrow$ 583.80. Probability change: rare disease 0.1128 $\rightarrow$ 0.9996.

*Takeaway.* Our counterfactual is centered on the diagnostically decisive signal and remains much more targeted. CFVAE barely changes the label, while REVISE flips it strongly but does so by globally amplifying multiple endocrine biomarkers.

**HCM vs ATTR.** **Original input:** SAM: 0; Low Left Ventricular Voltage: 0; Sarcomere Gene Mutation: 0; Family History: 0; Asymmetric Hypertrophy: 0; High Left Ventricular Voltage: 0; TTR Gene Mutation: 1; Amyloid Deposition: 1.

**Ours.**    Counterfactual output: SAM: $0 \to 0.9661$; Low Left Ventricular Voltage: $0 \to 0.9569$; Sarcomere Gene Mutation: $0 \to 0.8288$; Family History: $0 \to 0.7118$; Asymmetric Hypertrophy: $0 \to 0.6103$; High Left Ventricular Voltage: $0 \to 0.4346$; TTR Gene Mutation: $1 \to 0.8867$; Amyloid Deposition: $1 \to 0.8878$. Probability change: HCM $0.3361 \to 0.9017$.

**CFVAE.**    Counterfactual output: Asymmetric Hypertrophy: $0 \to 0.56$; Amyloid Deposition: $1 \to 0.15$; TTR Gene Mutation: $1 \to 0.83$; SAM, family history, voltage, and sarcomere features all rise only to about 0.14–0.17. Probability change: HCM $0.4684 \to 0.4833$.

**REVISE.**    Counterfactual output: Asymmetric Hypertrophy: $0 \to 0.56$; Amyloid Deposition: $1 \to 0.34$; TTR Gene Mutation: $1 \to 0.66$; SAM, family history, voltage, and sarcomere features all rise to about 0.28–0.33. Probability change: HCM $0.1274 \to 0.6426$.

*Takeaway.* Our counterfactual yields a stronger and more disease-specific shift toward HCM. Compared with CFVAE, REVISE captures a clearer manifold-like group change, but both baselines remain less decisive or more diffuse than our result.

## J. Extended Analysis of Unified Training

### J.1. Baseline Comparison on Pooled Datasets

In this section, we provide a more granular breakdown of the performance gap between our Unified DMAE and traditional discriminative baselines. Figure 7 illustrates the precision-recall trade-offs across the pooled 7-dataset cohort.

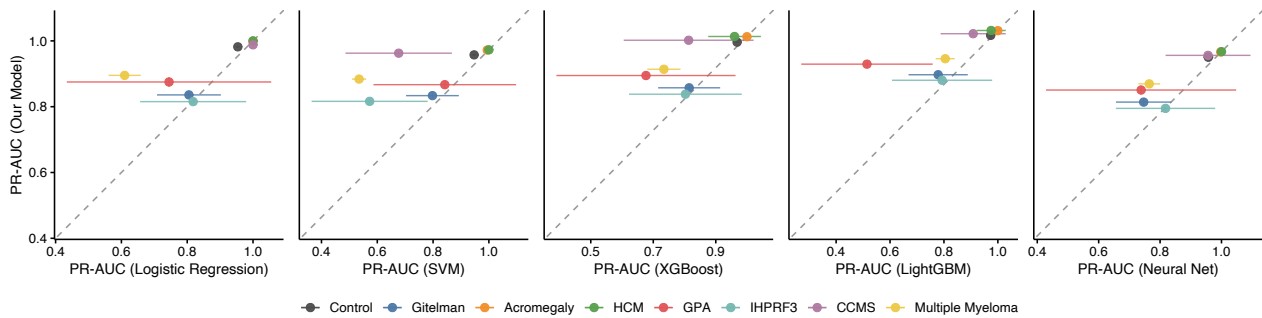

*Figure 7.* Performance comparison of different models on the pooled 7-dataset cohort. The Unified DMAE outperforms standard baselines in rare disease detection accuracy.

### J.2. Qualitative Case Studies of Counterfactuals

To further validate the clinical utility of the Unified DMAE, we present specific counterfactual examples generated from the pooled model in Table 8.

## K. Limitations

Our framework is intended as a reflective second-opinion tool rather than an autonomous diagnostic system, and three limitations should be made explicit. First, its usefulness depends on the quality of the learned latent representation and the AI predictor: under severe class imbalance, noisy records, or insufficient rare-disease coverage, the human-AI discrepancy signal may become weakened and the generated counterfactuals less trustworthy. Second, although the counterfactuals are designed to disrupt anchoring on common diagnoses, the prompts themselves could re-anchor clinicians toward the model's suggested rare alternative; we do not directly evaluate this risk, and dedicated clinician-in-the-loop studies are needed to quantify it. Third, the LLM-based evaluation in our study serves only as a scalable plausibility check and should not be read as a substitute for board-certified specialist review, which remains the authoritative source of clinical validation in this work.

*Table 8.* Clinical Evaluation of Multi-Domain Counterfactual Reasoning Cases.

| Scenario | Typical Cases (Feature Perturbations) | Prob. Shift | LLM Clinical Reasoning & Evaluation |
|---|---|---|---|
| **Scenario 1:** | **OGTT-GH_min:** nan $\rightarrow$ 118.43 
 **serum_GH:** 0.30 
 **IGF-1:** 174.00 | **Original:** 
 Rare: 0.2044 
 **CF:** 
 Rare: 0.9997 | The introduction of non-suppressible OGTT-GH (118.43 $\mu$g/L) provides definitive evidence of autonomous growth hormone secretion, a core indicator for Acromegaly. It corrects the misdiagnosis potentially caused by static markers, confirming the decisive role of dynamic functional tests. |
| **Scenario 2:** | **urine_potassium:** nan $\rightarrow$ 219.61 
 **serum_potassium:** nan $\rightarrow$ 3.17 
 **bicarbonate:** 27.60 
 **high_blood_pressure:** 0 
 **PH:** 7.47 | **Original:** 
 AI: Yes 
 Physician: No 
 **CF:** 
 AI: Yes 
 Physician: Yes | Significantly elevated urine potassium alongside low serum potassium indicates renal potassium wasting, a hallmark of Gitelman syndrome. This "low-blood, high-urine" pattern, combined with metabolic alkalosis and normal blood pressure, confirms the diagnosis. The simulation effectively demonstrates how filling key data gaps aligns physician judgment with AI predictions, mitigating anchoring bias. |
| **Scenario 3:** | **hemoptysis:** nan $\rightarrow$ 1 
 **PR3-ANCA:** nan $\rightarrow$ 1 
 **Age:** 59 
 **Sex_bool:** 1 
 **c-ANCA:** 0 
 **otitis media:** 0 | **Original:** 
 Rare: 0.48 
 **CF:** 
 Rare: 0.62 | The shift in PR3-ANCA and hemoptysis redirects the diagnostic logic from common respiratory infections to systemic vasculitis (GPA), outlining the inflammatory pathway. This case quantifies unobserved PR3-ANCA weights to provide a clear differential at the edge of uncertainty. |

