# OpenReview forum: "Beyond Accuracy: Latent Perturbations for Cognitive-Aware Diagnosis"
_ICML.cc/2026/Conference — ICML 2026 regular_

### Official Review · Reviewer_MjPq · 2026-02-16

**Soundness:** 2
**Presentation:** 3
**Significance:** 2
**Originality:** 2
**Overall Recommendation:** 4
**Confidence:** 3

**Summary:**

This paper proposes a DMAE-based framework for rare disease diagnosis. They design an objective to model the predictions of a clinician and the ground truth disease respectively. In the case of discrepancy between the two predictions, their framework enables perturbation-based counterfactual generation that highlights potential rare diseases. They validate their framework on 4 public datasets and 3 private datasets.

Claim: I have no experience in medical ML, hence I might miss something when it comes to related works and empirical study. I will focus on evaluating methodology/model.

**Compliance With Llm Reviewing Policy:**

Affirmed.

**Final Justification:**

The author's rebuttal has addressed most of my concerns.

**Key Questions For Authors:**

1. Introduction — the authors mentioned their method could help with the issue of limited data. Could the authors clarify how it helps specifically?
2. Section 2 — Could the authors explain how the overlapping symptom problem is addressed by their method?
3. Could the authors explain why perturbing $Z$ leads to plausible counterfactuals? Will the perturbed $Z$ be less meaningful or fall out of the manifold? I know some works would try to learn a more interpretable latent representation. In that case, perturbing certain features makes more sense. In contrast, I am not sure if perturbing $Z$ here is necessarily helpful.
4. Abstract - Could the authors clarify how their method mitigates the issue of “systematically under-considered relative to the observed evidence and learned diagnostic behavior.”

**Limitations:**

The paper could benefit from a more thorough discussion of the risks posed by poorly trained DMAE models and predictors.

**Strengths And Weaknesses:**

**Strength**
1. Rare disease diagnosis is an important problem to address in practice.
2. The paper is overall well written and easy to follow.

**Weakness**
1. I find the training objective proposed in Section 3 overly complicated. I am concerned that it will lead to difficulty in training. Besides, the proposed objectives are heuristics and there is no guarantee of getting a good latent representation.
2. Further, training requires access to both ground-truth labels and clinician diagnostic labels, which could be too limited in practice. The authors propose LR as a proxy in the appendix but I am concerned that will not be accurate enough for such a high-stake task.
3. Page 3 - the factorization in Section 3 is confusing. It seems like the authors assume $X$, $Y_{AI}$ and $Y_{Human}$ are independent of each other given $Z$. Could the authors clarify the validity of this assumption?
4. The authors do not provide a strong reason for choosing DMAE as the latent generative model. Why can't we use other generative models such as VAEs or diffusion models? Could the authors provide some theoretical justification or insights?

Please see **questions** for more issues.

---

> ### Author Rebuttal · Authors · 2026-03-31
>
> We sincerely appreciate Reviewer MjPq’s time and insightful feedback. Here is our response:
>
> ### W1. Role of loss terms and empirical support
>
> The multi-term objective is tailored for the rare-disease setting, specifically to make the AI aware of human decision regions and intentionally sensitive toward rare classes. **Core terms** ($L_{rec}, L_{AI}, L_{human}$) establish the diagnostic backbone, while **auxiliary terms** ($L_{contrast}, L_{gap}, L_{mask}$) explicitly model human-AI discrepancy to identify and counter common-disease anchoring. **Figure 6** confirms that removing any term degrades latent organization or rare-case sensitivity.
>
> ### W2. LR as a proxy for $Y_{Human}$
>
> LR simulates **heuristic cue-weighting** by emphasizing salient findings and under-weighting long-tail evidence [1,2]. As a **transparent, lower-capacity** model, it approximates anchor-prone human judgment based on a **limited set of key features**. When $Y_{human}$ is unavailable, it serves as a conservative proxy distinct from the model’s richer latent inference.
>
> [1] Marewski& Gigerenzer (2012). Heuristic decision making in medicine.
>
> [2] Wigton (1988). Use of linear models to analyze physicians' decisions.
>
> ### W3. Factorization assumption in Section 3
>
> The factorization is a **standard modeling assumption** in latent-variable generative models for **computational tractability**, not a claim of real-world clinical independence. We treat $Z$ as the latent patient state, and $X$, $Y_{AI}$, and $Y_{Human}$ as conditional readouts of that shared state. This is analogous to **VAE-style formulations** such as $p(z)p(x|z)p(y|z)$. We will clarify this approximation in Section 3.
>
> ### W4. DMAE vs. other generative models
>
> DMAE addresses the primary pain points of rare-disease diagnosis: **significant symptom overlap** and the frequent **neglect of non-routine features**. In small-cohort, structured settings with pervasive missingness, our masked denoising objective recovers latent structure from partial evidence—crucial because rare-disease diagnosis is often trivial only when features are complete. Compared to **Diffusion** (often less natural for sparse tabular data) or **VAEs** (which can be less stable in very low-data regimes), DMAE provides a robust backbone for bounded counterfactuals. Our goal is to highlight neglected diagnostic cues through controlled reconstruction from incomplete records, rather than high-fidelity unconditional generation.
>
> ### Q1. Handling limited data
>
> Our framework addresses rare-disease data limitations, especially incomplete records and extreme class imbalance, through: (1) **DMAE’s masked reconstruction** learns robust latent structure from partially observed data, turning missingness into a training signal; and (2) **focal loss, inverse-frequency weighting, and contrastive learning** reduce domination by common diagnoses and preserve rare-disease structure in the latent space. This improves data efficiency and predictive stability in sparse-evidence regimes, as supported by **Figure 3** under increasing imbalance ratios.
>
> ### Q2. Overlapping symptom problem
>
> Our framework addresses symptom overlap through: (1) **contrastive learning**, which increases rare–common separation in latent space; (2) the **cognitive-gap objective**, which reveals diagnostically relevant signals often under-considered by heuristic clinician judgment; and (3) **counterfactual probing via latent perturbation**, which identifies targeted evidence shifts needed to resolve ambiguity in overlapping cases.
>
> ### Q3. Plausibility of Latent Perturbations
>
> Perturbing $Z$ yields more plausible counterfactuals because it modifies a learned latent patient state rather than independently editing observed variables. Since clinical indicators are physiologically coupled, direct perturbations in $X$-space can create biologically inconsistent cases. In contrast, perturbing $Z$ and decoding back to $X$ changes correlated features jointly through the learned data distribution, preserving symptom dependencies. This process is further guided by the cognitive-gap objective and predefined clinical scenarios.
>
> ### Q4. "Under-considered" Evidence
>
>  “Systematically under-considered” refers to diagnoses plausible under observed evidence but under-prioritized by modeled clinician heuristics. Our framework mitigates this by jointly modeling disease distribution and human diagnostic behavior, using the cognitive gap to reveal diagnostic blind spots, and using counterfactuals to surface plausible but under-considered rare alternatives.
>
> ### Limitation
>
> If the DMAE or predictor is poorly trained, the latent space and counterfactuals may be unreliable. We mitigate this through validation, stabilized training, ablations, and specialist-audited constrained generation, but reliability still depends on model quality. We will make this limitation explicit and emphasize that the framework is intended as a second-opinion tool, not an autonomous diagnostic agent.

---

> > ### Author Rebuttal · Reviewer_MjPq · 2026-04-03
> >
> > Several of my concerns are not fully addressed:
> > 1. W1 - I understand the point of the objective. But my concerns about complexity and lack of theoretical guarantees remain.
> > 2. W2 - Simiarly, I understand why we use LR. But my concerns remain.
> > 3. W3 - I am concerned this conditional independence could be a less optimal inductive bias that might lead to bad representation.
> > 4. W4 - Could the authors provide some reference to support this argument?

---

> > > ### Author Response · Authors · 2026-04-04
> > >
> > > ### W1. Objective complexity and Guarantees
> > >
> > > Our support is empirical rather than theorem-based. The loss terms are not arbitrary heuristics but **functionally non-redundant**. In **Fig. 6**, removing reconstruction/contrastive/gap terms degrades latent organization, while removing task-specific terms degrades the corresponding predictor; for example, AI AUC drops from 0.96 to 0.89 and human AUC from 0.98 to 0.61. **This trend holds across all seven cohorts** in Table 7: removing AI loss lowers AI AUC to 0.86-0.98, while removing human or mask-related terms lowers human AUC to 0.61-0.98.
> > >
> > > Importantly, optimization is not performed by introducing all terms at once. As described in **Appendix C.2**, we use a **four-stage curriculum** (DMAE warm-up, AI predictor training, joint human-predictor/mask training, and fine-tuning), with **AUC variance below 0.02 across three random seeds**. This staged design suits rare-disease cohorts, which are small, structured, highly imbalanced, and often incomplete: the model **first** learns a reconstruction-based latent backbone from partial records and **then progressively adds** the human-modeling and rare/common-separation objectives.
> > >
> > > At the same time, we do **not** claim that the same multi-term design is universally necessary or optimal for **other** data regimes; it is **specifically motivated by the distinctive characteristics of rare-disease diagnosis**.
> > >
> > > ### W2. Will LR be accurate enough for such a high-stakes task?
> > >
> > > **The issue is not whether LR matches an ideal clinician, but whether it captures the imperfect initial judgment state that Y_human is designed to represent.** In our formulation, Y_human is defined as “preliminary, potentially anchored diagnostic decisions formulated under incomplete information.” The goal is therefore not to model an ideal final expert diagnosis, but **a coarse, cue-limited human judgment state that can miss rare-disease evidence when key information is absent**.
> > >
> > > Of course, the preferred setting is to use real clinician-provided first-impression labels under incomplete information; when such labels are unavailable, LR is a practical and reasonable proxy, consistent with prior work on linear models of physician judgment and heuristic medical decision-making (Wigton, 1988; Marewski & Gigerenzer, 2012).
> > >
> > > ### W3. Factorization assumption
> > >
> > > The factorization is meant to model **separate initial decision pathways** for AI and clinicians under a shared latent patient state, rather than to assert real-world conditional independence. This separation is central to our goal: to make the discrepancy between **comprehensive AI inference** and **cue-limited human judgment** visible, especially in settings where anchoring can cause clinicians to fixate early on a common diagnosis. In other words, we do **not** model an “AI-first-then-human” or “human-first-then-AI” pipeline; we model two parallel decisions from the same underlying case so that disagreement can be identified and used for counterfactual support. To avoid an overly restrictive inductive bias, the two heads are asymmetric and coupled with an explicit cognitive-gap objective, so the latent representation is encouraged to preserve systematic human–AI differences rather than collapse to an average label. Empirically, we do not observe such collapse: the latent-space analysis (**Appendix Fig. 5a–c**) shows structured disease clustering, meaningful attention patterns, and uncertainty regions consistent with this design.
> > >
> > > ### W4. DMAE and References
> > >
> > > **Learning from partially observed evidence (DMAE)**
> > >
> > > [1] He et al., 2022 (CVPR): masked autoencoding shows reconstruction from partial observations is an effective and scalable representation-learning paradigm.
> > >
> > > [2] Wu et al., 2023 (ICLR): DMAE combines masking and stochastic corruption to improve robustness and generalization.
> > >
> > > **Clinical missingness**
> > >
> > > [3] Liu et al., 2023 (Artif. Intell. Med.): systematic review of deep-learning-based handling of missing healthcare data.
> > >
> > > [4] Pereira et al., 2022 (IEEE JBHI): VAE-based partial multiple imputation for MNAR healthcare data.
> > >
> > > **Latent generative models for clinical counterfactuals**
> > >
> > > [5] Nagesh et al., 2023 (CHIL / PMLR 209): physician-facing clinical counterfactuals via a VAE-style framework.
> > >
> > > [6] Joshi et al., 2019: latent-space counterfactual recourse for actionable explanations.
> > >
> > > We use DMAE because rare-disease diagnosis is a partially observed inference problem: key evidence is often missing at the initial encounter, since relevant tests are non-routine and usually ordered only after suspicion arises. Masked autoencoders support reconstruction from partial observations [1], and DMAE combines masking and stochastic corruption to improve latent robustness [2]. This is well matched to incomplete rare-disease records and to clinically plausible counterfactual reasoning, consistent with prior work on healthcare missingness [3,4] and physician-facing clinical counterfactuals [5,6].

---

### Official Review · Reviewer_Qizq · 2026-03-09

**Soundness:** 2
**Presentation:** 4
**Significance:** 3
**Originality:** 3
**Overall Recommendation:** 4
**Confidence:** 2

**Summary:**

Clinicians in practice often settle for common instead of rare diagnoses due to a bias called "cognitive anchoring". Hence, the authors propose to challenge this initial assumption by providing a tool that creates a counterfactual from a common to a also possible rare analysis. The counterfactuals are created by perturbing the latent space of a denoising masked autoencoder.

**Compliance With Llm Reviewing Policy:**

Affirmed.

**Key Questions For Authors:**

1) There is a typo in "Latent Sp\textbf{x}ace" in figure 1 on the left side - it's no real problem for me, but you should maybe fix it before publication
2) AIs are notoriously bad at correctly deciding rare diseases, e.g., in histopathology [1], but also in radiology. In fact, rare diseases are often mentioned as the primary reason why radiologists are still necessary. Where does the assumption come from that one can compare human and AI predictions to detect rare diseases? Moreover, if AIs are so good in classifying rare diseases, why is the human still needed at all?


[1] Dippel, Jonas, et al. "Ai-based anomaly detection for clinical-grade histopathological diagnostics." Nejm Ai 1.11 (2024): AIoa2400468.

**Limitations:**

There are no limitations discussed, but it would be nice if there were, especially highlighting the assumptions about the base capabilities of the underlying AI systems.

**Strengths And Weaknesses:**

Pros:
+ creative application of counterfactuals
+ seems to perform well on selected tasks
+ well written paper
+ figure 1 explains the entire paper quite well

Cons:
- i do not understand the assumption that AIs are good at rare disease classification

---

> ### Author Rebuttal · Authors · 2026-03-31
>
> We sincerely thank Reviewer Qizq for recognizing the originality of our work and the clarity of our presentation. We especially appreciate your critical reflection on the base capabilities of AI in rare-disease settings, as it helps us sharpen the framing of our contribution.
>
> ### Q1. Typo in Figure 1
>
> Thank you for catching this. We will correct “Latent Spxace” to “Latent Space” in the revised manuscript.
>
> ### Cons/Q2. AI in rare-disease classification and why the human is still needed
>
> We do not assume that AI is a universally superior diagnostic authority for rare diseases, nor do we position it as a replacement for clinicians. Our claim is narrower: in this setting, AI and clinicians have **different strengths and different failure modes**, and our framework is designed to leverage that complementarity.
>
> More specifically, the AI branch is **intentionally sensitized toward rare diseases** during training, rather than optimized only for overall accuracy. By learning rare-disease structure in sparse, imbalanced settings, and through our specialized objective that sharpens common-versus-rare decision boundaries, it gains a relative diagnostic advantage in detecting plausible rare-disease alternatives. Through our discrepancy-based design, it is also explicitly aware of the human decision region—where clinicians may remain anchored to familiar common diagnoses. Its role is therefore not just to output a label, but to detect when the full evidence supports a plausible rare alternative that may otherwise be overlooked. In this sense, the framework goes **beyond accuracy**: stronger discrimination is the foundation, but its main value is to surface clinically meaningful disagreement and generate counterfactual “what-if” prompts that broaden the hypothesis space and support reflective reasoning.
>
> At the same time, the human remains essential because diagnosis is not only a pattern-recognition problem. Even when AI helps surface plausible rare-disease alternatives, it may still generate **unsafe suggestions**, especially if followed without human review. Clinicians therefore contribute forms of judgment that the model does not have: they can interpret findings in light of broader clinical context, incorporate information not fully captured in the structured record, screen out spurious or low-value suggestions, assess whether a proposed follow-up test is realistic and justified, and balance risks, costs, patient burden, and expected benefit. In this sense, the clinician’s role is not to duplicate the AI’s pattern recognition, but to **contextualize, filter, and serve as the final gatekeeper**.
>
> We will revise the manuscript to make this framing explicit. The framework is intended as a **reflective decision-support tool**, not an automated diagnostic system: AI helps broaden the space of plausible hypotheses through counterfactual generation, while the clinician remains responsible for plausibility screening and final judgment. We also agree with the reviewer’s limitation point and will clarify that the usefulness of this mechanism depends on the quality of the learned latent representation and AI branch; if the predictor fails to capture reliable rare-disease structure, the discrepancy signal and generated counterfactuals may be less trustworthy and less clinically informative.
>
> ### Limitation
>
> The usefulness of our framework depends more on whether the learned latent representation and predictor branches capture reliable rare-disease structure and produce clinically meaningful discrepancies. If the DMAE or predictor fails to represent such structure well—especially under severe imbalance, missingness, or noisy records—then the discrepancy signal may become **less informative**, and the generated counterfactuals may be **less trustworthy or clinically plausible**.
>
> We will therefore add a dedicated limitation discussion clarifying that the framework’s value lies in its clinical utility as a reflective decision-support mechanism, and that this utility depends on **the quality and calibration of the learned representation and predictors**. We will also cite the reviewer’s suggested reference [1] in this discussion.

---

> > ### Author Rebuttal · Reviewer_Qizq · 2026-04-01
> >
> > I am no real expert on this, but I would be fine to have the paper published at ICML.

---

> > > ### Author Response · Authors · 2026-04-02
> > >
> > > We sincerely thank you for your time and thoughtful evaluation. We truly appreciate your support for the publication of our paper at ICML and your recognition of its contribution to the challenging problem of rare-disease diagnosis.

---

### Official Review · Reviewer_M5Do · 2026-03-12

**Soundness:** 4
**Presentation:** 4
**Significance:** 4
**Originality:** 4
**Overall Recommendation:** 5
**Confidence:** 4

**Summary:**

This paper presents a framework aimed at mitigating "cognitive anchoring" in the diagnosis of rare diseases. The proposed method utilises a Denoising Masked AutoEncoder (DMAE) to learn latent representations of patient symptoms. By jointly training an AI predictor and a simulated "human" predictor, the model identifies cognitive gaps where human attention might overlook rare but plausible conditions. The system generates counterfactual clinical profiles by perturbing the latent space in directions that maximise the uncertainty of the simulated human clinician. The framework is evaluated across four public and three private rare-disease datasets, demonstrating improved label flip rates and lower perturbation sizes compared to baselines like REVISE and CFVAE.

**Compliance With Llm Reviewing Policy:**

Affirmed.

**Final Justification:**

After seeing the rebuttal, I realize the methodology parts are really plausible and practical. Also, the expert evaluation demonstrates that board-certified specialists found approximately 90% of the reviewed cases to be clinically plausible and useful, which strongly reinforces the real-world potential of the paper. I believe this approach is highly valuable and represents an important contribution to the challenging field of rare disease diagnosis, effectively addressing the non-trivial task of surfacing diagnostically meaningful alternatives.

**Key Questions For Authors:**

1. Can you rigorously justify using a Logistic Regression model as a proxy for complex human cognitive biases (like anchoring) in the public datasets? Have you validated that the LR model actually makes the same types of errors as real clinicians in those specific cohorts, or have some existing papers demonstrated it?
2. Regarding the expert evaluation, could you provide the exact number of cases evaluated by the board-certified specialists? Furthermore, what percentage of the model-generated counterfactuals were rated as "clinically plausible" versus "implausible"?
3. The perturbation bound $\epsilon$ is mentioned to be constrained by physiological reference intervals. How exactly is this enforced during the latent space perturbation phase, given that the perturbation happens in the latent space $Z$ rather than the raw feature space $X$?

**Limitations:**

yes

**Strengths And Weaknesses:**

### Strengths:
1. The idea of incorporating a cognitive-aware mechanism is innovative, and the paper explicitly models the cognitive anchoring effect that may arise in real-world diagnosis for rare diseases.
2. The paper presents various experiments on both public and private rare-disease datasets. It also includes counterfactual comparisons with REVISE and CFVAE, and Table 1 reports higher flip rates and lower RMSE.
3. Using a sparse self-attention mask to simulate human cognitive bandwidth constraints and identify divergence between AI and clinician focus is an interesting architectural design.
### Weaknesses:
1. The rationale for the chosen $Y^{human}$ is unclear. There are no provided labels for $Y^{human}$ in the public datasets, and the paper does not sufficiently justify the use of logistic regression to generate them. In addition, the claim that the attention mask corresponds to human focus needs further discussion.
2. The paper lists guiding clinicians toward alternative hypotheses as one of its contributions. However, it does not provide experiments showing that the proposed model actually improves clinicians’ diagnostic accuracy.
3. The use of RMSE to measure the proximity of counterfactuals is standard for continuous data. However, several datasets use binary or categorical features. Using RMSE on categorical multi-hot vectors is less interpretable.

---

> ### Author Rebuttal · Authors · 2026-03-31
>
> We sincerely thank Reviewer M5D0 for the careful and constructive feedback. We are encouraged that you found the cognitive-aware framing, multi-dataset evaluation, and sparse attention design meaningful. Below we respond to the main concerns.
>
> ### W1/Q1. The LR Proxy for $Y_{human}$ and the Attention Mask
>
> The linear form of LR simulates **heuristic cue-weighting** by emphasizing salient findings while under-weighting long-tail evidence, **consistent with prior work on medical decision-making under uncertainty [1,2]**. As a **transparent, lower-capacity** model, it approximates the way human judgment often reaches an initial diagnosis based on **a limited set of key features**. We do not claim that LR fully reproduces the complexity of real clinician biases in these cohorts; rather, when explicit $Y_{human}$ labels are unavailable, it provides a transparent and conservative approximation that preserves a clear distinction between anchor-prone, human-like judgment and the model’s richer latent inference.
>
> Likewise, this use of attention/masking as a computational approximation of selective attention is **consistent with prior work linking neural attention mechanisms to human selective attention [3,4]**. The sparse attention mask is not meant to directly measure true human attention, but to operationalize selective diagnostic focus in the human-simulation branch: the model predicts from a masked latent code $\tilde Z = m \odot Z$, and sparsity encourages reliance on only a subset of latent dimensions. Thus, the mask should be understood as an interpretable approximation of selective cue utilization.
>
> [1] Marewski & Gigerenzer (2012). Heuristic decision making in medicine.
>
> [2] Wigton (1988). Use of linear models to analyze physicians' decisions.
>
> [3] Fu et al. (2020). What can computational models learn from human selective attention? A review from an audiovisual unimodal and crossmodal perspective.
>
> [4] Guo et al. (2022). Attention mechanisms in computer vision: A survey.
>
> ### W2/Q2. Clinical utility claims and expert evaluation
>
> Successfully flipping common-to-rare disease states in sparse, imbalanced settings is itself non-trivial and indicates that the model can surface diagnostically meaningful rare-disease alternatives under realistic difficulty. We view this as evidence of the framework’s potential utility for reflective reasoning in complex cases.
>
> For the expert evaluation, board-certified specialists reviewed **30 cases** (full results in the anonymous supplementary PDF: [Supplementary PDF](https://osf.io/m3sf4/files/kxdhj?view_only=cf51e2fab0474714933f598758ca5d80)), of which approximately **90\%** were rated clinically plausible and useful. The remaining **10\%** were not judged clinically implausible, but rather of **low marginal utility** for highly experienced experts when the diagnostic reasoning was already relatively straightforward. We will report these numbers, the case-selection criterion, and the aggregate outcomes more clearly in the revision.
>
> ### W3. On RMSE for discrete features
>
> We adopt RMSE as a **unified proxy for overall perturbation magnitude** primarily to maintain **benchmark consistency** with established baselines (e.g., REVISE, CFVAE), which utilize similar distance-based proximity objectives. In this sense, RMSE serves as a common proxy for the scale of modification rather than a complete measure of semantic or clinical similarity. Your suggestion enhances the interpretability of our evaluation; to provide a more comprehensive comparison, we will supplement RMSE with feature-type-aware metrics, such as Hamming distance and $L_0$ sparsity, in the revised manuscript.
>
> ### Q3. How physiological bounds are enforced
>
> The two constraints operate in different spaces. (1) During counterfactual search, $\|\Delta Z\| \le \epsilon$ controls the direction and locality of the latent perturbation, so the search does not move too far from the original representation. (2) Physiological validity is then enforced in the decoded feature space: after decoding $Z+\Delta Z$ into $X_{cf}$, we apply differentiable penalties whenever reconstructed clinical variables fall outside physiologically admissible ranges, allowing gradients to steer the perturbation back toward clinically valid regions.
>
> In addition, because the DMAE decoder is trained on real-world clinical data, the latent-to-feature mapping is biased toward plausible clinical configurations. Only counterfactuals that pass the decoded-space physiological validity check are retained as final outputs. We will revise the manuscript to make this distinction and enforcement mechanism explicit.

---

> > ### Author Rebuttal · Reviewer_M5Do · 2026-04-02
> >
> > Thank you for the comprehensive rebuttal, which has completely resolved my concerns. I particularly appreciate the inclusion of the expert evaluation. Demonstrating that board-certified specialists found approximately 90% of the reviewed cases to be clinically plausible and useful strongly reinforces the real-world potential of your framework. I believe this approach is highly valuable and represents an important contribution to the challenging field of rare disease diagnosis, effectively addressing the non-trivial task of surfacing diagnostically meaningful alternatives.
> >
> > My only concern left is a practical one regarding the clinical workflow: it remains an open question whether busy clinicians will actually be willing to read a lengthy LLM-generated evaluation after making their initial diagnostic decision. However, I recognize that this is a deployment challenge that falls outside the methodological scope of this paper, and it does not detract from your core technical contributions. I really appreciate the effort put into this rebuttal, and I believe this paper is worth a higher rating. I have increased my score accordingly.

---

> > > ### Author Response · Authors · 2026-04-02
> > >
> > > We sincerely thank you for the thoughtful follow-up and for increasing the score. We also appreciate the remaining workflow concern.
> > >
> > > In rare-disease settings, clinicians may be more receptive to **targeted prompts for reconsideration**, since these cases often involve high uncertainty, overlapping symptoms, and diagnostic delay. In practice, the clinician-facing output could be a **concise structured prompt**, with any LLM-generated explanation or evaluation provided only as **optional supporting detail**: **(i) possible overlooked rare disease, (ii) key missing feature or recommended follow-up test, and (iii) why this evidence could materially shift the diagnosis**. This high-signal format would let busy clinicians quickly identify the main diagnostic pivot, with fuller explanation available only if needed. We agree that evaluating such **workflow-compatible presentation formats** is an important direction for future deployment. We are grateful again for your constructive engagement and recognition of the paper’s contribution.

---

### Official Review · Reviewer_seAx · 2026-03-16

**Soundness:** 2
**Presentation:** 4
**Significance:** 4
**Originality:** 4
**Overall Recommendation:** 5
**Confidence:** 4

**Summary:**

The work proposes a framework to assist clinicians in overcoming cognitive bias in the form of anchoring in clinical diagnosis, which leads to difficulties in diagnosing rare diseases.
To do so, the framework formulates a balanced ERM objective that uses both a
simulated cognitive bias and true rare diseases to build a latent representation that separates common disease from rare ones using a Denoising Masked Autoencoder.
The latent representation is then used to generate counterfactual synthetic patients on incomplete data,
which can provide valuable information for clinicians in not dismissing rare diseases.
The quality of counterfactuals for three specific patients is reviewed by both
clinical experts, as well as an LLM. Furthermore, the performance for
generating counterfactuals is compared to two baselines across seven rare
disease datasets.

**Compliance With Llm Reviewing Policy:**

Affirmed.

**Final Justification:**

I am very happy to see the full list of expert-reviewed examples, which fully addresses my initial main concern. Including the list in the appendix and clarifying these better in the manuscript will make the manuscript much stronger. I am still not convinced of the usefulness of the LLM-based evaluations, but I am okay with including them as supplemental to the much stronger empirical evidence.

All of my concerns have been addressed. Importantly, I am much more convinced of the soundness having seen the full list of expert reviews, which was my justification to recommend rejection of this work. As this is not an issue anymore, and the work tackles an important problem, is well written, and provides sufficient evidence, I change my recommendation to "Accept".

**Key Questions For Authors:**

1. My main concern with this work is with the lack of evidence for the claim of
   scientific soundness and clinical relevance of the method's counterfactuals.
   The evaluation of the method by doctors adds serious value to the
   qualitative analysis of the method and is the correct way to tackle this.
   However, reviews for only three patients with no notion of limits to the
   method (all reviews are praising the method) is not enough evidence.
   Additionally, there is no direct comparison to the counterfactuals produced
   by other methods.

2. As I understand it, both the "Flip" metric and the RMSE to indicate
   proximity of the counterfactual in Table 2 are more sanity checks that the
   counterfactual actually produce the intended "what-if" labels and examples.
   While it is worrying that some baselines do not produce counterfactuals with
   the correct labels, it appears to me that that should be a requirement of
   the method.

3. What insights do the LLM-Based evaluations provide in 5.1.2? Especially
   along true expert feedback, these do not seem particularly useful.

4. The results in Table 1 suggest that REVISE performs better than CFVAE
   (Nagesh et al., 2023), but in the CFVAE paper, the opposite is empirically
   shown. Why is there a discrepancy?

5. How do the latent embeddings in Figure 4 show that the method performs well?
   I cannot see a clear separation between rare and common diseases, as it
   appears a lot of common cases overlap with rare ones. Is this not what the
   framework was intended to solve? If it cannot do so fully, maybe it would be
   easier to compare it to a baseline experiment without the contrastive loss?
   Similarly, in the embeddings in the 3D plots in Figures 5 and 6, rare
   diseases are indistinguishable from the common ones, as they appear randomly
   distributed within the same groups as the common diseases.


### Less important questions:

6. With respect to Use-Case 2 in line 316 (left): how can one be sure that the
  model does not bias the clinician to some wrong diagnosis rather than promote
  their own critical thinking?

7. As far as my complete layperson knowledge with respect to rare diseases goes:
  many (some?) rare diseases can often be quite confidently detected by a single (usually expensive) test.
  I imagine it could easily happen that the model is biased to provide
  counterfactuals based on training examples where the "first-guess"
  was unrelated to the actual definitive final test.
  Adding a single such marker would definitely be a very small perturbation,
  but equate to just randomly conducting a test. Is this an issue, and if so,
  how do you make sure this does not happen? An appropriate control experiment
  with clinical experts would provide some evidence (I understand this is very
  hard to obtain).

**Limitations:**

The work does not discuss limitations. One possible limitation could be that
the framework could lead to biasing clinicians to simply go with the presented
counterfactual results rather than promoting critical thinking.

**Strengths And Weaknesses:**

### Strengths

1. The introduction is superbly written, and really explains and motivates the issue.
2. The issue that the work attempts to solve is immensely important.
3. The overall writing is very clear.
4. The expert reviews of the quality of the proposed counterfactuals is the
   absolute best way to validate the framework.
5. The supplementary provide ablation studies, implementation details, and
   further information on datasets and related work.

### Weaknesses

1. The main claim of the work, to provide a "scientifically sound" and
   "clinically relevant" framework to counter the cognitive bias of anchoring in
   clinical diagnosis, is not backed by sufficient evidence.
2. Expert reviews are limited to only three patients.
3. The baseline results disagree with prior work.

---

> ### Author Rebuttal · Authors · 2026-03-31
>
> We sincerely thank Reviewer seAx for the careful and constructive feedback. We are encouraged by your positive comments on the importance of the problem, the clarity of the presentation, and the evaluation methodology. Below we address your concerns in detail and clarify the corresponding revisions.
>
> ### W1/W2/Q1. Main concern: Evidence of Clinical Utility & Scale of Expert Evaluation
>
> 1. The expert-reviewed examples **were not limited to three patients**. As stated in Section 5.1.2, Figure 2 “**summarizes representative cases**” across three clinical scenarios owing to space constraints. Specialists reviewed 30 cases (full results at the anonymous supplementary PDF: [Supplementary PDF](https://osf.io/m3sf4/files/kxdhj?view_only=cf51e2fab0474714933f598758ca5d80)) with approximately 90% rated clinically plausible. The 10% with "low marginal utility" for senior experts were often clinically intuitive, consistent with our Impact Statement's focus on resource-constrained settings.
>
> 2. Our clinical validity is primarily grounded in the large-scale quantitative evaluation of **3,934** instances across seven heterogeneous datasets (Table 1), where performance is measured directly against clinical ground-truth labels. Figure 2 is included to illustrate what the generated counterfactuals look like in the three scenarios and provide clinically interpretable evidence beyond Flip/RMSE.
>
> 3. Figure 2 omits baseline cases because REVISE/CFVAE only perform generic label flipping, whereas we support scenario-conditioned counterfactuals. For a direct comparison, we added 9 matched label-flipping examples at [Supplementary PDF](https://osf.io/n2r89/files/3zgwm?view_only=570767820576472b8fede671660e9f71), demonstrating more targeted and clinically coherent results than baselines.
>
> ### Q2/Q3. Role of Flip/RMSE and LLM-Based Evaluation
>
> We report them because they capture the two core quantitative properties of a valid counterfactual: **Flip** measures whether the intended diagnosis change is achieved, and **RMSE** measures whether that change is made with relatively small perturbations. These remain important in rare-disease settings, where **extreme class imbalance and missing non-routine measurements are central diagnostic bottlenecks**, making valid counterfactual generation particularly challenging.
>
> While Flip and RMSE are necessary, they do not fully establish clinical usefulness. We therefore incorporate expert review for rigorous clinical validation. However, since board-certified specialists are a **scarce and costly resource** , the LLM-based evaluation serves as a **supplementary, scalable check** of plausibility, diagnostic relevance, and interpretability. We will make this distinction clearer in the revision.
>
> ### W3/Q4. REVISE vs. CFVAE discrepancy relative to the CFVAE paper
>
> We do not view the difference as a contradiction, because CF-VAE and our work study substantially **different tasks, data modalities, and overall evaluation protocols**: CF-VAE evaluates ICU time-series intervention prediction, whereas our setting is rare-disease diagnosis on heterogeneous tabular/phenotype data; CF-VAE also notes that REVISE often fails to converge in its setting.
>
> ### Q5. The role of latent-space visualizations
>
> Figures 4–6 illustrate the latent structure supporting counterfactual perturbations rather than claiming perfect linear separation. In real-world rare-disease diagnosis, substantial overlap is clinically expected due to shared nonspecific symptoms and incomplete evidence. **Quantitative results (Fig. 3, Tab. 2, App. Tab. 7) provide a more precise assessment** of model behavior than **low-dimensional** visualizations, including the experiment that removes contrastive loss in Fig. 6-(1). We will clarify this evidence hierarchy in the revision.
>
> ### Q6/Limitation. Use-case 2 and potential anchoring bias
>
> In Use-Case 2, the counterfactual is intended to make AI–clinician disagreement explicit by showing what bounded, clinically plausible evidence changes would alter the model’s decision. It is meant to **prompt reconsideration**, not to recommend a diagnosis or replace clinical judgment. We acknowledge that such a tool may itself introduce anchoring bias, which we do not directly evaluate here. This calls for clinician-in-the-loop evaluation in future work and we will make it explicit in revision.
>
> ### Q7. On "Shortcut" Counterfactuals
>
> Most of our datasets (e.g., Gitelman, MM, CCMS, IHPRF3) rely only on phenotype-level or routine features, precluding "shortcut" markers. For cohorts with specialized tests, our DMAE perturbs latent representations to reconstruct full counterfactual profiles, ensuring coherent clinical shifts across related symptoms rather than isolated flips. Specialist review also provides partial reassurance here.

---

> > ### Author Rebuttal · Reviewer_seAx · 2026-04-04
> >
> > I thank the authors for these detailed clarifications. I am very happy to see the full list of expert-reviewed examples, which fully addresses my initial main concern. Including the list in the appendix and clarifying these better in the manuscript will make the manuscript much stronger. I am still not convinced of the usefulness of the LLM-based evaluations, but I am okay with including them as supplemental to the much stronger empirical evidence.
> >
> > All of my concerns have been addressed. Importantly, I am much more convinced of the soundness having seen the full list of expert reviews, which was my justification to recommend rejection of this work. As this is not an issue anymore, and the work tackles an important problem, is well written, and provides sufficient evidence, I change my recommendation to "Accept".

---

> > > ### Author Response · Authors · 2026-04-04
> > >
> > > We sincerely thank you for your careful consideration, thoughtful follow-up and for changing your recommendation to Accept. We are very grateful that the full list of expert-reviewed examples resolved your main concern and strengthened your confidence in the soundness of the work.

---

### Decision · Program_Chairs · 2026-04-30

**Decision:**

Accept (regular)

**Comment:**

Reviewers reached consensus that this paper should be included in the ICML proceedings. There was substantial back-and-forth between reviewers and authors during the rebuttal phase; however, almost all of the issues raised by reviewers were adequately addressed. In revising the paper, authors must include the full list of expert-reviewed examples in the appendix and clarify the findings in the manuscript.